# PARAMETER EFFICIENT MULTI-TASK MODEL FUSION WITH PARTIAL LINEARIZATION

**Anke Tang[1], Li Shen[2]\*, Yong Luo[1]\*, Yibing Zhan[2], Han Hu[3], Bo Du[1], Yixin Chen[4], Dacheng Tao[5]**
[1]Wuhan University, China [2]JD Explore Academy, China [3]Beijing Institute of Technology, China
[4]Washington University, USA [5]Nanyang Technological University, Singapore
[1]{anketang,luoyong,dubo}@whu.edu.cn
[2]mathshenli@gmail.com,zybjy@mail.ustc.edu.cn [3]hhu@bit.edu.cn
[4]chen@cse.wustl.edu [5]dacheng.tao@ntu.edu.sg

## ABSTRACT

Large pre-trained models have enabled significant advances in machine learning and served as foundation components. Model fusion methods, such as task arithmetic, have been proven to be powerful and scalable to incorporate fine-tuned weights from different tasks into a multi-task model. However, efficiently fine-tuning large pre-trained models on multiple downstream tasks remains challenging, leading to inefficient multi-task model fusion. In this work, we propose a novel method to improve multi-task fusion for parameter-efficient fine-tuning techniques like LoRA fine-tuning. Specifically, our approach partially linearizes only the adapter modules and applies task arithmetic over the linearized adapters. This allows us to leverage the the advantages of model fusion over linearized fine-tuning, while still performing fine-tuning and inference efficiently. We demonstrate that our partial linearization technique enables a more effective fusion of multiple tasks into a single model, outperforming standard adapter tuning and task arithmetic alone. Experimental results demonstrate the capabilities of our proposed partial linearization technique to effectively construct unified multi-task models via the fusion of fine-tuned task vectors. We evaluate performance over an increasing number of tasks and find that our approach outperforms standard parameter-efficient fine-tuning techniques. The results highlight the benefits of partial linearization for scalable and efficient multi-task model fusion.

## 1 INTRODUCTION

Pre-trained models play a crucial role in machine learning systems, serving as foundational components. In order to optimize their performance for specific downstream tasks (Ilharco et al., 2022; Wortsman et al., 2022b; Matena & Raffel, 2022), address biases or undesired behavior (Santurkar et al., 2021; Ribeiro & Lundberg, 2022; Murty et al., 2022), align them with human preferences (Ouyang et al., 2022; Ribeiro & Lundberg, 2022), or incorporate new information (Mitchell et al., 2022a;b), it is often necessary to further customize or edit these models after pre-training.

Multi-task model fusion is a powerful approach to extracting knowledge from models fine-tuned on different downstream tasks, allowing us to create a unified model that performs well across multiple tasks. This approach proves to be helpful when only the fine-tuned model can be obtained but the data remains private and inaccessible (Wu et al., 2019; Lou et al., 2020; Tang et al., 2023); besides, it can expedite the fine-tuning of the multi-task model, since compared to the pre-trained model, we have a better starting point from which to fine-tune (Kaddour, 2022; Sanyal et al., 2023). In recent studies, researchers have introduced many powerful methods for editing pre-trained models and merging task-specific fine-tuned models. We further introduce this field in Section 2.

The vast parameter size of pre-trained models poses challenges in terms of computational efficiency and memory usage during the fine-tuning process; these difficulties further lead to inefficient multi-task model fusion. Fine-tuning large-scale models requires significant computational resources and

---

*Corresponding authors.

memory, making the process inefficient. To address this concern, many parameter-efficient fine-tuning (PEFT) techniques are proposed, these approaches significantly reduce the number of parameters that need to be fine-tuned, meanwhile achieving comparable performance to full parameter fine-tuning. However, naively combining models that were fine-tuned in a parameter-efficient manner can more readily result in representational interference between tasks, which makes model fusion algorithms suboptimal. While some research has explored fusing parameter-efficient fine-tuned models for multi-task model fusion (Chronopoulou et al., 2023; Zhang et al., 2023; Huang et al., 2023), performance still lags considerably behind fusing fully fine-tuned models. Therefore, the key challenge is performing PEFT while also preventing negative interference between task-specific representations. Motivated by these concerns, we aim to enhance the multi-task model fusion capabilities of parameter-efficient fine-tuned models.

In this work, we present a novel approach to improve the multi-task fusion capability of parameter-efficient fine-tuning models. Recent advances in understanding task arithmetic and weight disentanglement have demonstrated that linearizing the entire model and fine-tuning the corresponding tangent model in tangent space can enable more effective task arithmetic (Guillermo Ortiz-Jimenez et al., 2023). While promising, completely linearizing a large pre-trained model can be computationally expensive. Typically, this approach requires two to three times the computational resources needed for fine-tuning and inference. Our key insight is that we can perform efficient fine-tuning and disentangle task representations by only linearizing a subset of parameters appended to a fixed pre-trained backbone. In essence, we are proposing a hybrid approach that leverages parameter-efficient fine-tuning for efficiency, while locally linearizing the adaptable modules to attain enhanced disentanglement and improved multi-task fusion capabilities.

Our experiments on image classification and natural language processing tasks demonstrate that our partial linearization technique enables more effective model fusion, achieving superior performance across tasks compared to conventional PEFT methods and model fusion algorithms alone. In some cases, our proposed method is even comparable to full fine-tuning. In addition to the direct comparison of multi-task model fusion performance, we also visualize the weight disentanglement gain of our method on different downstream task pairs. The results show that our method can effectively improve the weight disentanglement of parameter-efficient fine-tuning models, which is the key to improving the multi-task fusion capability of parameter-efficient fine-tuning models.

To summarize, our contributions are as follows:

- We propose a novel partial linearization method for parameter-efficient fine-tuning models in order to improve the multi-task fusion capability of fine-tuned task-specific models with a low computational cost overhead.
- We apply our method to the LoRA modules to construct Linearized LoRA (L-LoRA) modules and conduct extensive experiments on seven tasks from the GLUE benchmark to demonstrate that our method is effective in improving the multi-task fusion capability of fine-tuned task-specific models.
- We present an extension of weight disentanglement property and weight disentanglement error for parameter-efficient fine-tuning models to analyze the impact of the linearization process on parameter-efficient modules. We evaluate fine-tuned models to visualize and analyze the weight disentanglement gain of L-LoRA on downstream tasks.

## 2 RELATED WORK

**Model Fusion**. In practical training scenarios, it is common practice to train multiple models with various hyperparameter configurations. Subsequently, the top-performing individual model is selected, or an ensemble of models is constructed to attain optimal performance (Dietterich, 2000). However, deploying an ensemble of models incurs additional computing expenses during inference. Meanwhile, selecting a single top-performing model typically entails discarding the remaining trained models, resulting in neglecting their encoded knowledge.

Model fusion integrates the knowledge from multiple models into a single unified model (Li et al., 2023). There are two main works for model fusion. The first line of work focuses on the fusion of entire models. Interpolating the entire weight space of multiple models by taking their average or weighted combination can be effective for model fusion as demonstrated in previous work (Garipov

et al., 2018; Ilharco et al., 2022; Matena & Raffel, 2022; Wortsman et al., 2022a; Ilharco et al., 2023; Liu & Soatto, 2023). When the models are not well aligned or lie in different loss basins, techniques have been proposed to align the feature representation before fusion (Liu et al., 2022a; Wang et al., 2022; Ainsworth et al., 2023; Jin et al., 2023). This feature alignment helps matching the models to similar behaviors or transforming to a same loss basin. Fusing the full model can leverage knowledge from all layers but can be computationally expensive.

The second line of work focuses on fusing parts of the model. For example, (George Stoica et al., 2023) aligns and fuses a large portion of the models to combine models trained on disjoint tasks and supports partial zipping up to a specified layer to create a multi-head model. (Yadav et al., 2023) removes redundant and potentially interfering values and resolves sign conflicts before doing weight interpolation. Other researches explore merging parameter-efficient modules as a more lightweight approach (Chronopoulou et al., 2023; Huang et al., 2023; Zhang et al., 2023; Wu et al., 2023).

**Parameter-efficient fine-tuning (PEFT)**. The use of transfer learning from pre-trained models has become a widely accepted practice, resulting in impressive performance across various domains (Devlin et al., 2019; Radford et al., 2019). In addition to fine-tuning all the model parameters, parameter-efficient fine-tuning techniques have been proposed to reduce the number of parameters that need to be fine-tuned, such as adapter tuning (Houlsby et al., 2019), prefix tuning (Li & Liang, 2021), prompt tuning (Lester et al., 2021), LoRA (Hu et al., 2021) and (IA)$^3$ Liu et al. (2022b) *et al*. These methods only update a small subset of (extra) parameters, lowering computational costs and memory requirements compared to full fine-tuning, while retaining comparable performance.

## 3 RETHINKING MODEL FUSION IN PEFT SETTING

In this section, we first introduce the preliminary formulas of parameter-efficient fine-tuning and define task vectors for parameter-efficient fine-tuning models. Then, we introduce the weight disentanglement property and weight disentanglement error in PEFT setting.

### 3.1 PRELIMINARY

Under the framework of supervised learning, assume we have a set of tasks $T = \{\tau_1, \tau_2, \cdots, \tau_n\}$ and a pre-trained language model $f_{\theta_0}$ parameterized by $\theta_0$ which is trained on a massive dataset. Each task $\tau_i$ is associated with a dataset $D_{\tau_i} = \{(x_{\tau_i}^{(j)}, y_{\tau_i}^{(j)})\}_{j=1}^{s_i}$, where $s_i$ is dataset size of $D_{\tau_i}$, $\{x_{\tau_i}^{(j)}\}_{j=1}^{s_i}$ is the input data, and $\{y_{\tau_i}^{(j)}\}_{j=1}^{s_i}$ is the output data.

Given a specific task $\tau_i$, the pre-trained model $f_{\theta_0}$ is fine-tuned on the task-specific dataset $D_{\tau_i}$ to obtain a task-specific model. We consider full parameter fine-tuning and parameter-efficient fine-tuning methods. If the model $f_{\theta_0}$ is fully fine-tuned on task $\tau_i$, the parameters $\theta_0$ are updated to $\theta_i$ by minimizing the loss function $\theta_i = \arg\min_\theta \mathcal{L}(\tau_i; \theta)$ on the task-specific dataset $D_{\tau_i}$. Otherwise, if the model $f_{\theta_0}$ is parameter-efficiently fine-tuned on task $\tau_i$, the original parameters $\theta_0$ are fixed and some extra added parameters $\phi$ (the number of parameters of $\phi$ is much less than $\theta_0$, i.e., $|\phi|/|\theta_0| \ll 1$) are updated to $\phi_i$ by minimizing the loss function $\phi_i = \arg\min_\phi \mathcal{L}(\tau_i; \theta_0, \phi)$ on the task-specific dataset $D_{\tau_i}$. Where $\mathcal{L}$ is typically the Cross-Entropy loss function to maximize conditional posterior probability $p(y|x)$, more specifically

$$\mathcal{L}(\tau_i; \theta_0, \phi) = \frac{1}{s_i} \sum_{j=1}^{s_i} \mathcal{L}_{\text{CE}}\left(f_{\theta_0}\left(x_{\tau_i}^{(j)}; \phi\right), y_{\tau_i}^{(j)}\right) \in \mathbb{R}^+. \tag{1}$$

**Task vector**. We define the task vector as the difference between the fine-tuned *trainable* parameters and their initial state. The task vector is denoted as $\nu_i = \theta_i - \theta_0$ for full fine-tuning or $\nu_i = \phi_i - \phi_0$ for parameter-efficient fine-tuning. We elaborate on this definition in Appendix A, discussing the rationale behind this definition and its associated benefits.

### 3.2 WEIGHT DISENTANGLEMENT FOR PARAMETER-EFFICIENT FINE-TUNING

In this subsection, we discuss and present an extension of the weight disentanglement property and weight disentanglement error to parameter-efficient fine-tuning models, which are original introduced in (Guillermo Ortiz-Jimenez et al., 2023).

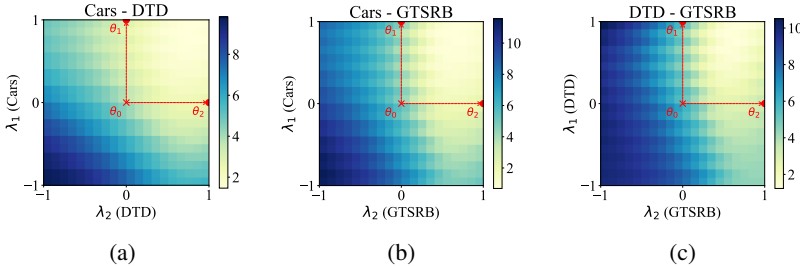

Figure 1: **Loss landscape visualization**. Here, we visualize the loss landscape $\mathcal{L}(\tau_1; \theta) + \mathcal{L}(\tau_2; \theta)$ for CLIP model on combinations of three downstream image classification tasks by interpolating on the 2D plane. $\theta = \theta_0 + \sum_{i=1}^{2} \lambda_i (\theta_i - \theta_0)$, where $\theta_0$ are the pre-trained weights, $\theta_i$ are the task-specific full fine-tuned weights for task $\tau_i$. From these heatmaps, we observe that task-specific models reside in the same loss basin when evaluated on the joint task.

**Weight disentanglement for parameter-efficient fine-tuning**. The weight disentanglement is defined by the function outputs of the merged model. Consider a PEFT model denoted as $f_{\theta_0}(x; \phi_0)$ whose tunable weights are initialized as $\phi_0$. We define this model to possess weight disentanglement characteristics in relation to a specific set of task vectors $\{\nu_i | \nu_i = \phi_i - \phi_0\}_{i \in [n]}$ and their corresponding support datasets $\{D_{\tau_i}\}_{i \in [n]}$ if the following condition is satisfied:

$$f_\theta\left(x; \phi + \sum_{i=1}^{n} \lambda_i \nu_i\right) = \sum_{i=1}^{n} g_i(x; \lambda_i \nu_i) + g_0(x), \tag{2}$$

where $g_i(x; \lambda_i \nu_i) = 0$ for $x \notin D_{\nu_i}$ and $i = 1, 2, ..., n$ and $g_0(x) = 0$ for $x \in \bigcup_{i \in [n]} D_{\tau_i}$. This condition ensures that the model $f_\theta(x; \phi)$ can be expressed as a linear combination of individual terms $g_i(x; \lambda_i \nu_i)$, incorporating the respective task vectors and an additional term $g_0(x)$. Through adhering to this disentanglement condition, the model demonstrates the desired capability of effectively separating and disentangling the weights associated with the function outputs, enhancing its capacity to capture task-specific information.

However, it is important to highlight that weight disentanglement is a characteristic specific to the function outputs and is not directly linked to their performance. In other words, a model may exhibit weight disentanglement property yet still perform weakly. Since weight disentanglement of function outputs does not guarantee task success in itself, as other factors, such as the evaluation metrics, can be non-linear to determine the overall performance. Therefore, it is crucial to consider weight disentanglement alongside other performance metrics when analyzing model behavior and effectiveness across different tasks.

**Weight disentanglement error in PEFT setting**. Given two task vectors $\{\nu_i\}_{i=1,2}$, we define the extension of *disentanglement error* for model $f_{\theta_0}(x; \phi_0)$ as follows:

$$\xi(\lambda_1, \lambda_2) = \sum_{i=1}^{2} \mathbb{E}_{x \sim P(D_{\tau_i})} \left[\text{dist}(f_{\theta_0}(x; \phi_0 + \lambda_i \nu_i), f_{\theta_0}(x; \phi_0 + \lambda_1 \nu_1 + \lambda_2 \nu_2))\right]. \tag{3}$$

Where $\text{dist}(\cdot, \cdot)$ represents any chosen distance metric between two vector outputs. Taking classification tasks as an example, dist can be chosen as the prediction error, i.e., $\text{dist}(y_1, y_2) = \mathbb{1}(y_1 \neq y_2)$. Intuitively, this measures how much the prediction changes when adding each task vector individually versus adding both together. If task knowledge is well-disentangled, adding the specialized vectors independently or together should yield similar predictions, minimizing $\xi$. The disentanglement error quantifies the degree of interference between task vectors. Lower values indicate task knowledge that can be combined with less destructive interaction during model merging.

### 3.3 GEOMETRIC INTUITION FOR MODEL FUSION

In previous work, it is generally believed that the closer the orthogonal between task vectors, the less interference between tasks and the better the effect of weight disentanglement (Ilharco et al., 2023; Chen et al., 2023; Guillermo Ortiz-Jimenez et al., 2023). The intuition is that orthogonal

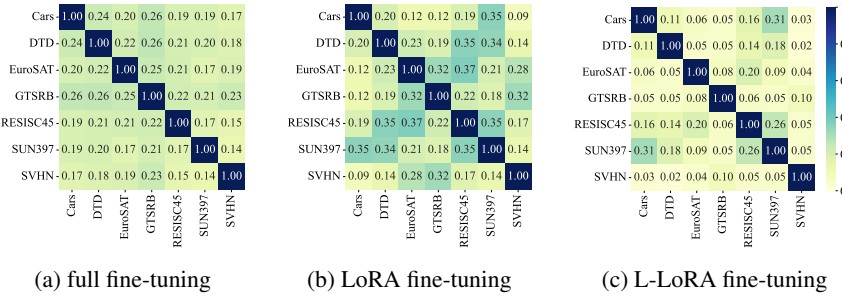

(a) full fine-tuning  (b) LoRA fine-tuning  (c) L-LoRA fine-tuning

Figure 2: **Similarity heatmaps**. These figures show heatmaps of the cosine similarity between task vectors from task-specific CLIP models (Radford et al., 2021) fine-tuned on different tasks. (a) Cos similarity matrix of task vectors when using full fine-tuning of the entire model. (b) Task vector similarities when using LoRA. (c) Cos similarity of task vectors when using L-LoRA, our proposed partial linearization approach that linearizes PEFT modules and fine-tunes in tangent space.

task vectors indicate that the specialized knowledge captured for each task lies in distinct subspaces with minimal overlap or redundancy. This enables better preservation of specialized knowledge and avoids destructive interference during model merging.

We visualize the loss landscape of the joint tasks in Figure 1 (Li et al., 2018), where we interpolate between the pre-trained weights $\theta_0$ and two task-specific fine-tuned weights $\theta_1, \theta_2$ for CLIP models (Radford et al., 2021). The 2D heatmaps show the loss values $\mathcal{L}(\tau_1, \theta) + \mathcal{L}(\tau_2, \theta)$ evaluated on the joint tasks. Notably, we observe that the task-specific models reside at the edge of the same low-loss basin, with no significant barriers or discontinuities in between. It provides geometric intuition for why a simple linear arithmetic operation of task-specific parameters $\theta_1$ and $\theta_2$ can produce effective merging for multitask learning, as empirically observed.

The results in Figures 2(a-b) and 8(a-b) show the cosine similarity between task vectors from CLIP and Flan-T5 (Chung et al., 2022), which are fully fine-tuned or LoRA fine-tuned on image classification tasks and natural language processing (NLP) tasks respectively. We observe that vectors from full fine-tuning are closer to orthogonal than those from LoRA, which indicates that models fine-tuned with full fine-tuning are more independent than those in LoRA. This finding is consistent with the discussion about task addition in (Ilharco et al., 2023), the experimental results from Figures 4 and 5 also support our statement. The experimental details are described in Appendix C.

**Remark 3.1** *At first glance, it may seem unfair to compare the task vectors from full fine-tuning to those from parameter-efficient fine-tuning methods, since full fine-tuning has access to much more trainable parameters. In fact for full fine-tuning we have $f_\theta(x) = f_\theta(x; \phi_0)$ so that*

$$\frac{\langle \nu_i, \nu_j \rangle}{\|\nu_i\|_2 \|\nu_j\|_2} = \frac{\langle [\theta_i - \theta_0, \mathbf{0}], [\theta_j - \theta_0, \mathbf{0}] \rangle}{\|[\theta_i - \theta_0, \mathbf{0}]\|_2 \|[\theta_j - \theta_0, \mathbf{0}]\|_2} = \frac{\langle [\theta_i - \theta_0, \phi_0 - \phi_0], [\theta_j - \theta_0, \phi_0 - \phi_0] \rangle}{\|[\theta_i - \theta_0, \phi_0 - \phi_0]\|_2 \|[\theta_j - \theta_0, \phi_0 - \phi_0]\|_2};$$

*on the other hand, for parameter-efficient fine-tuning we have*

$$\frac{\langle \nu_i, \nu_j \rangle}{\|\nu_i\|_2 \|\nu_j\|_2} = \frac{\langle [\mathbf{0}, \phi_i - \phi_0], [\mathbf{0}, \phi_j - \phi_0] \rangle}{\|[\mathbf{0}, \phi_i - \phi_0]\|_2, \|[\mathbf{0}, \phi_j - \phi_0]\|_2} = \frac{\langle [\theta_0 - \theta_0, \phi_i - \phi_0], [\theta_0 - \theta_0, \phi_j - \phi_0]}{\|[\theta_0 - \theta_0, \phi_i - \phi_0]\|_2 \|[\theta_0 - \theta_0, \phi_j - \phi_0]\|_2}.$$

*Therefore, the comparison between full fine-tuning and parameter-efficient fine-tuning methods can be made fair by viewing them as updating different subsets of the joint parameter space $(\theta, \phi)$.*

## 4 METHODOLOGY

Based on the above observations, we propose a novel partial linearization method for parameter-efficient fine-tuning models in order to enhance the weight disentanglement and improve the multi-task fusion capability of fine-tuned task-specific models with a low computational cost overhead.

### 4.1 LINEARIZED PARAMETER-EFFICIENT FINE-TUNING

To monitor the progressive of trainable parameters $\phi$ throughout the training trajectory, we treat $\phi$ as a function of the training time $t$, denoted as $\phi(t)$. Where t ranges from 0 to $T$. By employing a

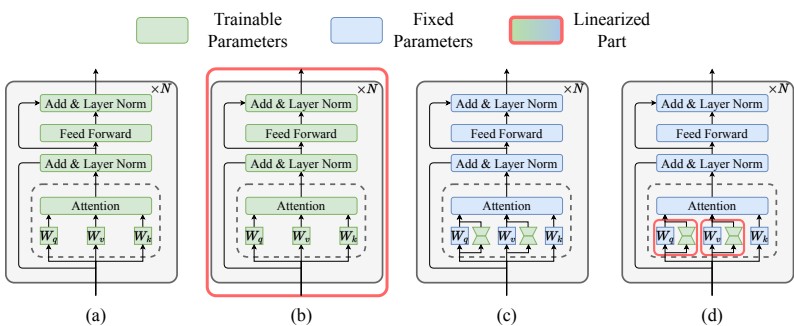

Figure 3: **Four types of fine-tuning paradigms**. (a) Full parameter fine-tuning. (b) Full-model linearization. (c) Parameter-efficient fine-tuning. (d) Linearized parameter-efficient fine-tuning. In this paper, we explore LoRA fine-tuning and linearized LoRA (L-LoRA) fine-tuning.

first-order Taylor expansion, we can streamline the dynamics of network learning as (Weng, 2022):

$$f_{\theta_0}(x; \phi(t)) \approx f_{\theta_0}^{\text{lin}}(x; \phi(t)) = f_{\theta_0}(x; \phi(0)) + \nabla_\phi f_{\theta_0}(x; \phi(0))^\top (\phi(t) - \phi(0)). \quad (4)$$

The linearized model, described by Eq.(4) also referred to as a tangent model, approximates the behavior of the neural network $f_{\theta_0}(x, \phi(t))$ at a specific time $t$, where $\theta_0, f_{\theta_0}(x; \phi(0)), \nabla_\phi f_{\theta_0}(x; \phi(0))$ are all constants. Consequently, the linearized function $f_{\theta_0}^{\text{lin}}(\phi(t); x)$ is a linear function of $\phi(t)$.

Grounded in the key result detailed in Appendix B, where we characterize the evolution of model outputs. We hypothesize that by partially linearizing only a subset of parameters within the adaptable modules, the model can gain from a more disentangled representation. This partial linearization facilitates the separation of task-specific knowledge; hence, model fusion becomes more robust against the negative transfer effects and task interference in multi-task learning settings.

Figure 3 illustrates four different types of fine-tuning paradigms. The first three are existing methods, while the fourth is our proposed partial linearization fine-tuning method. (a) The full fine-tuning paradigm, all parameters $\theta$ are updated during fine-tuning. (b) The full-model linearization paradigm, but instead we fine-tune the tangent model in the tangent space. It is worth noting that although the Jacobian-vector products can be computed in a single forward pass (Pearlmutter, 1994), training and inference in this paradigm are usually twice or three times as expensive as full fine-tuning, a quantitative comparison is shown in Tables 3 and 4. (c) The PEFT paradigm, only a small number of parameters $\phi$ are updated while $\theta$ is fixed at $\theta_0$. (d) the linearized PEFT (L-PEFT) paradigm, which is similar to PEFT fine-tuning, where $\phi$ is updated while $\theta_0$ is fixed. However, in L-PEFT, only the linearized PEFT modules are fine-tuned in the tangent space. This approach incurs only a fraction of the training and inference costs compared to standard PEFT fine-tuning. In this paper, we explore LoRA fine-tuning and linearized LoRA (L-LoRA) fine-tuning.

From Figures 2 and 8, we observe that task vectors from L-LoRA fine-tuning are closer to orthogonal than those from LoRA, which indicates that models fine-tuned with L-LoRA are more task-independent than those in LoRA. This is because the trainable parameters $\phi$ with L-LoRA are fine-tuned in the tangent space, which is a linear space, while $\phi$ in LoRA are fine-tuned in the original ambient space, which is a non-linear space. Besides, in some cases, the cosine similarity of task vectors from full fine-tuning are more orthogonal than those from L-LoRA. This may due to the larger space of trainable parameters available to full fine-tuning, providing an extremely high-dimensional space to encode specialized task knowledge. So even though L-LoRA tunes parameters in a linear subspace, the total capacity is restricted compared to full fine-tuning.

## 4.2 THE PARAMETER-SCALING LAW FOR WEIGHT DISENTANGLEMENT

Previous findings in (Ilharco et al., 2023; Guillermo Ortiz-Jimenez et al., 2023) also suggest an empirical parameter-scaling law wherein weight disentanglement improves with increased number of trainable model parameters. Intuitively, (1) The larger the number of parameters, the stronger the expressive ability of the model, which can better learn the correlation between different tasks, thus facilitating weight disentanglement. (2) The over-parameterization brought about by a large number of parameters can provide more degrees of freedom to learn the feature representation required for

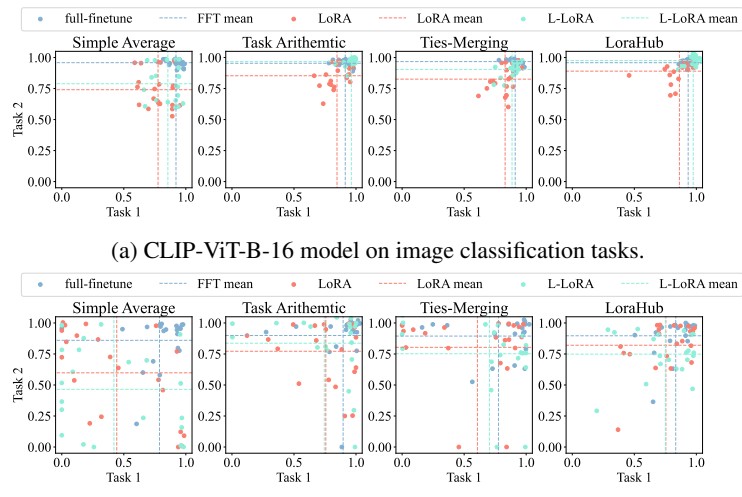

(a) CLIP-ViT-B-16 model on image classification tasks.

(b) Flan-T5-Base model on NLP tasks.

Figure 4: **Pairs of model fusion**. These figures show scatter plots demonstrating the performance of different model fusion techniques on pairs of tasks. Each plot corresponds to a different fusion method. The x and y axes in each plot denote the normalized scores on the two tasks. Points indicate the performance of specific instances. Dashed lines represent the average performance per task for each method. (a) Image classification tasks. (b) NLP tasks.

each task and reduce parameter competition or interference between different tasks. Figure 5(a) shows that even though L-LoRA fine-tuning has far fewer trainable parameters compared to full fine-tuning (<1%), it achieves comparable average normalized scores to full fine-tuning.

## 5 EXPERIMENTS

In this section, we conduct experiments on diverse tasks spanning both vision and natural language domains. We employ CLIP and Flan-T5 models on vision and language domains, respectively. For detailed descriptions, please refer to Appendix C.

Throughout the following experiments, we compare the performance when models are fine-tuned using full fine-tuning, LoRA (Hu et al., 2021), and our proposed L-LoRA fine-tuning. A more detailed experimental description of model fine-tuning is provided in Appendix C.

**Baseline methods**. We compare our fine-tuning method combined with four baseline multi-task model fusion methods: simple average, task arithmetic (Ilharco et al., 2023; Zhang et al., 2023), ties-merging (Yadav et al., 2023), and LoraHub (Huang et al., 2023). Please refer to Appendix E.1 for a more detailed description of these methods and hyper-parameter settings in our experiments.

### 5.1 MULTI-TASK MODEL FUSION ON VISION AND LANGUAGE DOMAINS

We extensively investigate the multi-task model fusion by considering all possible subsets of the tasks, amounting to a total of $2^n - (n+1)$ subsets (excluding subsets with a single task and an empty set). We employ various model fusion algorithms to accomplish this. Subsequently, we assess the performance of the final model across the downstream tasks in both the vision and language domains. Detailed experimental settings are provided in Appendix E.

Figure 4 shows scatter plots demonstrating the performance of four fusion methods: simple averaging, task arithmetic, ties-merging, and LoraHub. Each plot corresponds to one fusion approach applied to two tasks. This analysis of pairs of tasks highlights distinctions between fusion techniques. For image classification tasks, we observe that L-LoRA fine-tuning surpasses LoRA fine-tuning with all four model fusion algorithms, with task arithmetic and LoraHub even exceeds full fine-tuning. As for NLP tasks, we observe comparable results between LoRA fine-tuning and L-LoRA fine-tuning, although not as strong as full fine-tuning in some cases.

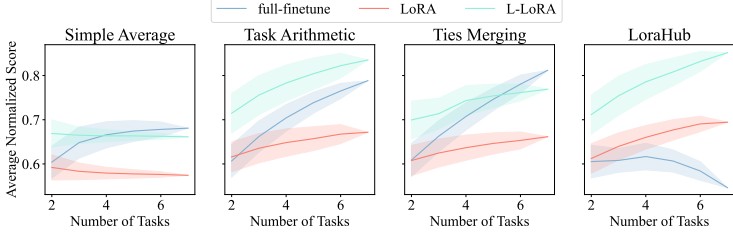

(a) CLIP-ViT-B-16 model on image classification tasks.

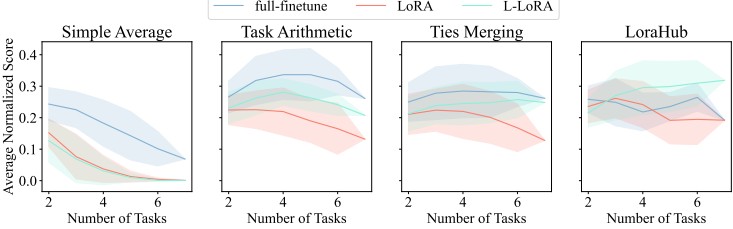

(b) Flan-T5-Base model on NLP tasks.

Figure 5: **Multi-task model fusion**. we construct multi-task models by utilizing task vectors specific to individual tasks, employing various model fusion algorithms. In the evaluation, the x-axis represents the number of task vectors used in building the multi-task model, while the y-axis represents the average normalized scores of the resulting models across all seven downstream tasks. The lines on the plot represent the average normalized scores of all multi-task models when considering a fixed number of tasks, and the shaded area corresponds to the standard deviation.

Table 1: The average of the normalized scores for all combinations (avg. $\pm$ std.).

| Fine-tuning method | Simple average | Task Arithmetic | Ties-merging | LoraHub |
|---|---|---|---|---|
| | | *CLIP-ViT-B-16* | | |
| full fine-tuning | $0.65 \pm 0.04$ | $0.69 \pm 0.06$ | $0.69 \pm 0.06$ | $0.61 \pm 0.03$ |
| LoRA fine-tuning | $0.58 \pm 0.18$ | $0.64 \pm 0.03$ | $0.63 \pm 0.03$ | $0.65 \pm 0.04$ |
| L-LoRA fine-tuning | $\mathbf{0.66} \pm 0.02$ | $\mathbf{0.77} \pm 0.05$ | $\mathbf{0.73} \pm 0.04$ | $\mathbf{0.77} \pm 0.05$ |
| | | *Flan-T5-Base* | | |
| full fine-tuning | $\mathbf{0.19} \pm 0.08$ | $\mathbf{0.32} \pm 0.08$ | $\mathbf{0.28} \pm 0.08$ | $0.24 \pm 0.06$ |
| LoRA fine-tuning | $0.06 \pm 0.07$ | $0.21 \pm 0.07$ | $0.21 \pm 0.08$ | $0.23 \pm 0.07$ |
| L-LoRA fine-tuning | $0.05 \pm 0.07$ | $0.26 \pm 0.05$ | $0.24 \pm 0.06$ | $\mathbf{0.28} \pm 0.08$ |

Figures 5, 10 and 11 show the performance of multi-task models constructed using different fine-tuning methods and fusion algorithms over an increasing number of task vectors. Table 1 shows the average over all combinations. In line with (Ilharco et al., 2023), we compare the normalized scores of merged multi-task models. The normalized score is defined as the absolute score divided by the score of the corresponding single-task model. These results highlight the capability of model fusion methods to effectively leverage knowledge into a single unified multi-task model.

In Figure 5, we also notice that for full fine-tuning, the ties-merging performs better than LoraHub on visual tasks, while this is not the case on NLP tasks. Ties-meriging introduces several innovations to address the challenges of parameter interference. This indicates that the degree of interference between visual tasks is higher than that between NLP tasks. Compare the cosine similarities matrix between Figure 2(a) and Figure 8(a), indeed, the task vectors on vision domain have higher cosine similarity, which is consistent with this model fusion phenomenon. Since the higher cosine similarity between task vectors implies greater redundancy and overlap in the knowledge captured. This results in more destructive task interference when naively merging the specialized models.

For both image classification tasks and NLP tasks, we observe that the normalized scores of L-LoRA surpass LoRA fine-tuning in most cases, and L-LoRA even exceeds full fine-tuning with task

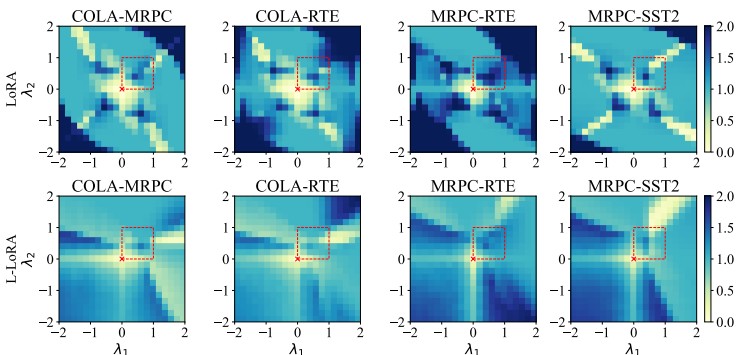

Figure 6: **Weight disentanglement error visualization** (LoRA vs L-LoRA): The heatmaps demonstrate the disentanglement error, denoted as $\xi(\lambda_1, \lambda_2)$, for both a LoRA fine-tuned flan-t5-base models and their L-LoRA counterparts across various NLP task pairs. In these heatmaps, lighter regions indicate lower weight disentanglement error, indicating stronger weight disentanglement. Within the heatmaps, the red box outlines the specific search space utilized to calculate the optimal $\lambda$ hyperparameters for our multi-model fusion experiments (refer to Section 5.1 and Appendix E).

arithmetic and ties-merging. In addition, LoRA is more effective than L-LoRA for a smaller number of tasks in some cases, but as the number of task vectors increases, L-LoRA surpasses LoRA. This phenomenon also suggests the effectiveness of L-LoRA fine-tuning for multi-task model fusion, we further discuss it in Appendix E.2.

A more in-depth analysis comparison between LoRA and L-LoRA is provided in Figures 10 and 11. For further elaboration and supplementary details, please refer to Appendix E.

## 5.2 WEIGHT DISENTANGLEMENT VISUALIZATION

To analyze the impact of the partial linearization process of PEFT modules on weight disentanglement, we evaluate LoRA fine-tuned models and their L-LoRA counterparts to visualize in Figure 6.

The heatmaps allow us to compare the error landscapes in the interpolation space. There are clear differences between the LoRA and L-LoRA heatmaps. The results indicate that the LoRA models exhibit a negligible disentanglement error confined to a narrow region in proximity to their pretrained model. On the contrary, the L-LoRA models showcase a low-error distribution across a wider area. These findings strongly suggest that the proposed partial linearization process employed in L-LoRA fine-tuning offers notable advantages in effectively disentangling the weights learned for each specific task, thereby enhancing its ability to capture task-specific information and improve performance across diverse tasks.

## 6 CONCLUSION

In this work, we propose a novel partial linearization method to improve multi-task fusion for parameter-efficient fine-tuning techniques. Specifically, our approach partially linearizes only the adapter modules and applies model fusion algorithms over the linearized adapters. This allows us to leverage the the advantages of model fusion over linearized fine-tuning, while still performing fine-tuning and inference efficiently. Our experiments are conduct on both vision and language domains extensively. The results demonstrate that partially linearized LoRA models enable a more effective fusion of multiple tasks into a unified multi-task model, outperforming standard LoRA fine-tuning and model fusion algorithms alone.

Future work could further explore partial linearization, where some parameters are fine-tuned in a linear space while others remain nonlinear. This hybrid approach could trade-off between specialized single-task performance and weight disentanglement capability.

ACKNOWLEDGMENTS

This work is supported in part by the National Key Research and Development Program of China under No. 2023YFC2705700, the Special Fund of Hubei Luojia Laboratory under Grant 220100014, the National Natural Science Foundation of China (Grant No. 62225113, U23A20318 and 62276195), and CCF-Zhipu AI Large Model Fund OF 202224.

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

## A DEFINITION OF TASK VECTOR

In this section, we provide insights into the rationale behind defining the task vector and the associated advantages. Our definition of the task vector for full fine-tuning is consistent with that introduced in (Ilharco et al., 2023). However, for parameter-efficient fine-tuning, we define the task vector as $\phi_i - \phi_0$ on the trainable parameter space instead of the change of $\theta$ on the original model parameter space as done in (Jiang et al., 2023; Zhang et al., 2023). This deliberate decision is carefully considered and corresponds to the nature of our fine-tuning approach.

Take LoRA as an example, when partially linearizing a LoRA model, we focus on the dynamics of the trainable parameters and compute the Jacobian-vector product for all trainable parameters, specifically the LoRA parameters denoted by $\phi$. The original model parameters, $\theta_0$, remain constant throughout the fine-tuning process and serve as fixed foundations or buffers that enable the adaptable components of the model, the LoRA parameters, to manifest their effects.

From a mathematical standpoint, we perform a first-order Taylor expansion exclusively on the trainable parameters. This expansion does not include the parameter merging operations of LoRA:

$$f_{\theta_0}^{\text{lin}}(x; \phi) = f_{\theta_0}(x; \phi_0) + \nabla_\phi f_{\theta_0}(x; \phi_0)^\top (\phi - \phi_0) \tag{5}$$

It is essential to emphasize that the application of this Taylor expansion is restricted to the parameters explicitly designated as trainable. These are the parameters that undergo active adjustments during the fine-tuning phase, representing the modifiable components of the model.

What's more, while our experiments have primarily utilized LoRA fine-tuning, such a definition proves beneficial for generalizing our method to other parameter-efficient fine-tuning techniques, such as adapter-tuning.

## B MODEL LINEARIZATION FROM A NTK PERSPECTIVE

In this section, we show that a linearized model is more capable to separate the function outputs in different directions in weight space, thus leading to a enhancement of weight disentanglement and offering advantages over non-linear models for model fusion.

The linearized model, described by Eq.(4) approximates the behavior of the neural network $f_{\theta_0}(x, \phi(t))$ at a specific time $t$, where $\theta_0, f_{\theta_0}(x; \phi(0)), \nabla_\phi f_{\theta_0}(x; \phi(0))$ are all constants. Consequently, the linearized function $f_{\theta_0}^{\text{lin}}(\phi(t); x)$ is a linear function of $\phi(t)$.

Assuming the trainable parameters are updated by gradient descent on task $\tau_i$, hence

$$\nabla_\phi \mathcal{L}(\tau_i; \theta_0, \phi) = \mathbb{E}_{(x_{\tau_i}, y_{\tau_i}) \sim D_{\tau_i}} [\nabla_\phi f_{\theta_0}(x_{\tau_i}; \phi) \nabla_f \mathcal{L}_{\text{CE}}(f_{\theta_0}(x_{\tau_i}; \phi), y_{\tau_i})] \tag{6}$$

$$\phi(\Delta t) - \phi(0) = -\eta \nabla_\phi \mathcal{L}(\tau_i; \theta_0, \phi(0)) \tag{7}$$

$$= -\eta \mathbb{E}_{(x_{\tau_i}, y_{\tau_i}) \sim D_{\tau_i}} [\nabla_\phi f_{\theta_0}(x_{\tau_i}; \phi(0)) \nabla_f \mathcal{L}_{\text{CE}}(f_{\theta_0}(x_{\tau_i}; \phi(0)), y_{\tau_i})], \tag{8}$$

where $\eta$ denotes the learning rate and $\mathcal{L}_{\text{CE}}$ is the cross-entropy loss. Consequently,

$$f_{\theta_0}^{\text{lin}}(x; \phi(\Delta t)) - f_{\theta_0}(x; \phi(0)) \tag{9}$$

$$= \nabla_\phi f_{\theta_0}(x; \phi(0))^\top (\phi(\Delta t) - \phi(0)) \tag{10}$$

$$= -\eta \mathbb{E}_{(x_{\tau_i}, y_{\tau_i}) \sim D_{\tau_i}} [\nabla_\phi f_{\theta_0}(x; \phi(0))^\top \nabla_\phi f_{\theta_0}(x_{\tau_i}; \phi(0)) \nabla_f \mathcal{L}_{\text{CE}}(f_{\theta_0}(x_{\tau_i}; \phi(0)), y_{\tau_i})] \tag{11}$$

$$= -\eta \mathbb{E}_{(x_{\tau_i}, y_{\tau_i}) \sim D_{\tau_i}} [\boldsymbol{K}(x, x_{\tau_i}; \phi(0)) \nabla_f \mathcal{L}_{\text{CE}}(f_{\theta_0}(x_{\tau_i}; \phi(0)), y_{\tau_i})], \tag{12}$$

where $\boldsymbol{K}$ is known as neural tangent kernel (Jacot et al., 2018), which is defined as $\boldsymbol{K}(x, x'; \phi) = \langle \nabla_\phi f_{\theta_0}(x; \phi), \nabla_\phi f_{\theta_0}(x'; \phi) \rangle \in \mathbb{R}^{|x|} \times \mathbb{R}^{|x|} \to \mathbb{R}^{|f| \times |f|}$. Since the NTK term is fixed throughout the training, the change in the model output $\Delta f_{\theta_0}^{\text{lin}}(x; \phi(\Delta t))$ is solely determined by the gradient of the loss $\nabla_f \mathcal{L}_{\text{CE}}(f_{\theta_0}(x_{\tau_i}; \phi(0)), y_{\tau_i})$. This establishes the connection between the neural network dynamics and kernel methods, allowing us to analyze the model through the lens of NTK.

**Connection between linearization and weight disentanglement.** Integrating the concept of a linearized model, we can express how the linearized model's output for a given input sample $x$ evolves from its initial state $\phi(0)$ over time as follows: For any small time increment $\Delta t$, the difference in

output from the linearized model $f_{\theta_0}^{\text{lin}}$ at the state $\phi(\Delta t)$ compared to the original model $f_{\theta_0}$ at the initial state $\phi(0)$ is calculated by projecting the gradient of the model output with respect to its initial parameters $\nabla_\phi f(x; \phi(0))$ onto the parameter change $\phi(\Delta t) - \phi(0)$. Symbolically, this relationship is denoted as:

$$f_{\theta_0}^{\text{lin}}(x; \phi(\Delta t)) - f_{\theta_0}(x; \phi(0)) = \nabla_\phi f_{\theta_0}(x; \phi(0))^\top (\phi(\Delta t) - \phi(0)), \tag{13}$$

which further simplifies to a term involving the expectation over the dataset $D_{\tau_i}$, capturing the interaction between the kernel representation $\boldsymbol{K}$ of the input samples and the gradient of the loss function $\nabla_f \mathcal{L}_{\text{CE}}$. Mathematically, this term is represented as Eq.(12):

$$-\eta \mathbb{E}_{(x_{\tau_i}, y_{\tau_i}) \sim D_{\tau_i}}[\boldsymbol{K}(x, x_{\tau_i}; \phi(0)) \nabla_f \mathcal{L}_{\text{CE}}(f_{\theta_0}(x_{\tau_i}; \phi(0)), y_{\tau_i})]. \tag{14}$$

Building upon this derivation, we arrive at a fundamental equation that depicts the proportional relationship between the expected task-specific gradient of the loss function and the modification in parameters for the linearized model:

$$\phi(\Delta t) - \phi(0) \propto \mathbb{E}_{(x_{\tau_i}, y_{\tau_i}) \sim D_{\tau_i}}[\boldsymbol{K}(x, x_{\tau_i}; \phi(0)) \nabla_f \mathcal{L}_{\text{CE}}(f_{\theta_0}(x_{\tau_i}; \phi(0)), y_{\tau_i})]. \tag{15}$$

In essence, this formula underscores the direct correspondence between the changes in weights of a linearized model and the gradient of the loss. It hints at the underlying mechanisms of weight disentanglement, revealing that the adjustment in parameters is essentially directed by the aggregated gradient information from the task-specific data. This understanding is crucial as it provides insight into how a model's weights evolve in response to the learning process.

## C    MODEL FINE-TUNING DETAILS

All of our experiments were performed using the same hardware consisting of eight NVIDIA GTX 3090 GPUs with 24GB video memory. We used PyTorch 2.0 and Python 3 throughout all experiments.

**Model, Evaluation Tasks, and Metrics**. We conduct experiments on diverse tasks spanning both vision and natural language domains. The model, evaluation datasets and metrics settings are as follows:

- For vision tasks, we utilize CLIP (Radford et al., 2021) as our pre-trained model. For fine-tuning, we employ the CLIP-ViT-B-16 models on seven image classification tasks derived with the same random seed 42 to initialize the parameter-efficient models. These tasks are Stanford Cars (Krause et al., 2013), DTD (Cimpoi et al., 2014), EuroSAT (Helber et al., 2018), GT-SRB (Stallkamp et al., 2012), RESISC45 (Cheng et al., 2017), SUN397 (Xiao et al., 2010), and SVHN (Netzer et al., 2021). We report top-1 accuracy as a performance metric.

- For NLP tasks, we utilize Flan-T5 (Chung et al., 2022) as our pre-trained language model. For fine-tuning, we employ the Flan-T5-base models on seven tasks derived from the GLUE benchmark (Wang et al., 2018) with the same random seed 42 to initialize the parameter-efficient models. These tasks are CoLA, MNLI, MRPC, QQP, RTE, SST2, and STSB. We report Spearman's $\rho$ for STSB and accuracy for others.

**Vision domain**. All the fine-tuning experiments on image classification tasks using AdamW optimizer. We fine-tune pre-trained CLIP-ViT-B-16 (Radford et al., 2021) downloaded from HuggingFace [1]. For image classification tasks we use the text prompt template 'a photo of {label}' to compute the text embeddings. We initialize all parameter-efficient modules (LoRA and L-LoRA) with the same random seed 42. We use a batch size of 64 and a learning rate of 1e-5 and a weight decay of 0.1 with a warm-up cosine scheduler for 6000 steps for all downstream tasks.

**Language domain**. The fine-tuning experiments on NLP tasks in our study follow the same experimental setup as described in (Raffel et al., 2020) using Adam optimizer. We fine-tune pre-trained Flan-T5-base model (Chung et al., 2022) download from HuggingFace [2]. In line with the previous setup, we convert all the downstream NLP tasks into a text-to-text format. For specific examples,

---

[1] https://huggingface.co/openai/clip-vit-base-patch16
[2] https://huggingface.co/google/flan-t5-base

Table 2: Hyperparameters for model fine-tuning.

| Scenario | Learning rate | Steps | Batch size | Weight decay | LoRA r |
|---|---|---|---|---|---|
| *CLIP-ViT-B-16 on vision tasks* | | | | | |
| full fine-tuning | 1e-5 (warm-up cos) | 6000 | 64 | 0.1 | - |
| LoRA | 1e-5 (warm-up cos) | 6000 | 64 | 0.1 | 16 |
| L-LoRA | 1e-5 (warm-up cos) | 6000 | 64 | 0.1 | 16 |
| *Flan-T5-Base on NLP tasks* | | | | | |
| full fine-tuning | 1e-5,2e-5 | 2000 | 16 | 0 | - |
| LoRA | 1e-5,2e-5,3e-5,4e-5 | 2000 | 16 | 0 | 16 |
| L-LoRA | 1e-5,2e-5,3e-5,4e-5 | 2000 | 16 | 0 | 16 |

Table 3: Summary of fine-tuning statistics.

| Fine-tune diagram | Trainable params. | Total params. | Trainable% | VRAM | Checkpoint size |
|---|---|---|---|---|---|
| *CLIP-ViT-B-16 on vision tasks (batch size=64)* | | | | | |
| full fine-tuning | 85M | 85M | 100% | $\approx$ 12.4GB | 328MB |
| LoRA (r=16) | 294K | 86M | 0.34% | $\approx$ 9.2GB | 1.2MB |
| L-LoRA (r=16) | 294K | 93M | 0.32% | $\approx$ 9.3GB | 1.2MB |
| *Flan-T5-Base on NLP tasks (batch size=16)* | | | | | |
| full fine-tuning | 247M | 247M | 100% | $\approx$ 16.3GB $\times$ 4 | 945MB |
| LoRA (r=8) | 885K | 248.5M | 0.36% | $\approx$ 11.3GB $\times$ 4 | 3.5MB |
| LoRA (r=16) | 1,769K | 249.3M | 0.71% | $\approx$ 11.4GB $\times$ 4 | 6.9MB |
| L-LoRA (r=8) | 885K | 291.8M | 0.30% | $\approx$ 11.3GB $\times$ 4 | 3.5MB |
| L-LoRA (r=16) | 1,769K | 293.5M | 0.60% | $\approx$ 11.4GB $\times$ 4 | 6.9MB |

please refer to F.1. This standardized format facilitates the training process and ensures consistency across different tasks, enabling effective fine-tuning of the models. We initialize all the parameter-efficient modules (LoRA and L-LoRA) with the same random seed 42. For standard full fine-tuning, a constant learning rate of 1e-5 or 2e-5 was used for 2000 steps with a batch size of 16 and no weight decay. For LoRA and L-LoRA fine-tuning, a sweep of learning rates from 1e-5 to 4e-5 was conducted, while keeping the other hyperparameters the same. The hyperparameter LoRA r is set to 16 for both LoRA and L-LoRA but is not applicable for full parameter fine-tuning. The learning rate sweep aims to find optimal values for different methods. Each fine-tuning run is conducted on four GPUs, utilizing the power of distributed data parallelism (DDP). This approach allows for efficient training of the model by dividing the batch across multiple GPUs and processing it simultaneously.

Table 2 summarizes the key hyperparameters used during the fine-tuning of CLIP-ViT-B-16 and Flan-T5-base model under three different scenarios: standard full fine-tuning (FFT), fine-tuning with LoRA, and fine-tuning with L-LoRA.

Table 3 presents a summary of the fine-tuning statistics for several different fine-tune diagrams. The table provides information on the trainable parameters, total parameters, trainable parameter percentage, VRAM utilization (with a batch size of 16) and checkpoint sizes.

Taking the Flan-T5-Base model as an example, full parameter fine-tuning updates all parameters, while leading to very high VRAM usage of around 16.3GB per GPU and a large 945MB checkpoint size. On the contrary, the Low-Rank adaptation (LoRA) models have significantly fewer trainable parameters, ranging from 885K to 1.7M. This corresponds to only 0.3-0.7% of the total parameters being trainable. Consequently, the VRAM usage is much lower at 11.3-11.4GB per GPU. The checkpoint sizes are also smaller, from 3.5-6.9MB. The linearized LoRA fine-tuning has VRAM usage similar to LoRA. They have more total parameters at 291.8-293.5M because we need to cache

Table 4: Summary of training and inference cost.

| Scenerio | Training (Adam) | | Inference | |
|---|---|---|---|---|
| | Time | VRAM | Time | VRAM |
| *CLIP-ViT-B-32 (batch size=64)* | | | | |
| full fine-tuning (Figure 3(a)) | 8.25 it/s | $\approx$ 4.2GB | 13.28 it/s | $\approx$ 0.9GB |
| full linearization (Figure 3(b)) | 4.94 it/s | $\approx$ 6.3GB | 8.52 it/s | $\approx$ 2.0GB |
| LoRA (r=16) (Figure 3(c)) | 8.43 it/s | $\approx$ 2.6GB | 13.01 it/s | $\approx$ 0.9GB |
| L-LoRA (r=16) (Figure 3(d)) | 7.55 it/s | $\approx$ 2.7GB | 10.65 it/s | $\approx$ 1.0GB |
| *CLIP-ViT-B-16 (batch size=64)* | | | | |
| full fine-tuning | 2.97 it/s | $\approx$ 12.4GB | 8.92 it/s | $\approx$ 1.6GB |
| full linearization | 1.49 it/s | $\approx$ 20.1GB | 2.50 it/s | $\approx$ 3.7GB |
| LoRA (r=16) | 3.83 it/s | $\approx$ 9.1GB | 8.60 it/s | $\approx$ 1.6GB |
| L-LoRA (r=16) | 3.24 it/s | $\approx$ 9.2GB | 7.23 it/s | $\approx$ 1.8GB |
| *CLIP-ViT-L-14 (batch size=16)* | | | | |
| full fine-tuning | 2.55 it/s | $\approx$ 13.9GB | 8.23 it/s | $\approx$ 1.8GB |
| full linearization | 1.32 it/s | $\approx$ 21.8GB | 2.26 it/s | $\approx$ 4.8GB |
| LoRA (r=16) | 3.49 it/s | $\approx$ 9.1GB | 7.99 it/s | $\approx$ 1.8GB |
| L-LoRA (r=16) | 2.99 it/s | $\approx$ 9.3$G$B | 6.55 it/s | $\approx$ 1.9GB |

the initiation values of LoRA modules to compute the Jacobian vector products. The checkpoint sizes match LoRA as well.

In summary of Table 3, full parameter fine-tuning has the most parameters but is costly to fine-tune. LoRA reduces the trainable subset of weights, enabling more efficient adaptation. L-LoRA increases the total parameters while maintaining a small trainable subset. Our partial linearization approach incurs minimal additional cost compared to standard LoRA fine-tuning. The training and inference costs increase only marginally and are negligible over LoRA.

Table 4 summarizes the training and inference cost for different fine-tuning methods applied to CLIP vision models of varying sizes. The costs are broken down in terms of training/inference time (iterations/sec) and video RAM (VRAM) usage.

Several key trends can be observed from Table 4. Full linearization is the slowest for both training and inference due to the increased computational cost. It also requires the most VRAM. LoRA is faster than full tuning for training and on par for inference, while using less VRAM for training due to the parameter-efficient fine-tuning. L-LoRA is slightly slower than LoRA. Its resource usage is comparable to LoRA.

In Figure 7, we report the single task performance of different fine-tuning methods: full parameter fine-tuning, LoRA fine-tuning and L-LoRA fine-tuning. These results demonstrate that nonlinear fine-tuning consistently achieves higher single-task performance compared to linearized fine-tuning. Among the non-linear methods, full fine-tuning gives the highest task-specific accuracy, while LoRA tuning performs slightly worse, but is more parameter efficient. L-LoRA fine-tuning exhibits the lowest individual task performance. However, it provides improved multitask fusion capabilities. Although L-LoRA underperforms on single tasks, when evaluated on joint tasks, it can surpass LoRA fine-tuning due to enhanced representation disentanglement. The trade-off between specialized single-task accuracy and generalizable multi-task performance is evident when comparing full fine-tuning to LoRA and L-LoRA in Section 5.1 and Appendix E.

Future work could further explore partial linearization, where some parameters are fine-tuned in a linear space while others remain nonlinear. This hybrid approach could trade-off between specialized single-task performance and weight disentanglement. Specifically, partitioning the parameters into fixed and linearly tuned subsets may combine the representation benefits of linearization with the expressiveness of nonlinear tuning. The non-linear parameters could capture task-specific adaptations, maximizing per-task accuracy. Meanwhile, the linearized parameters could encourage independence in different directions of weight space between tasks.

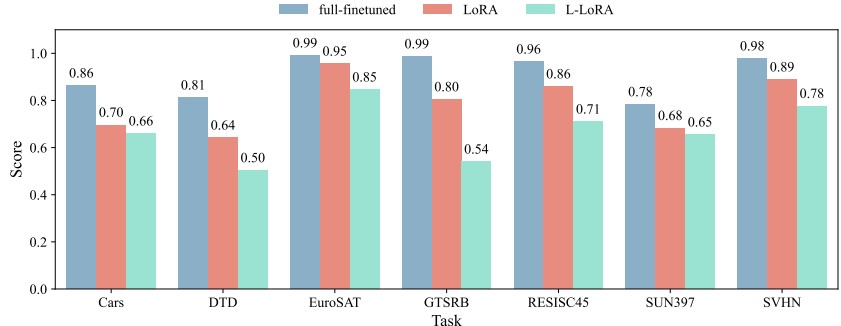

(a) Performance of task-specific CLIP-ViT-B-16 models on downstream image classification tasks.

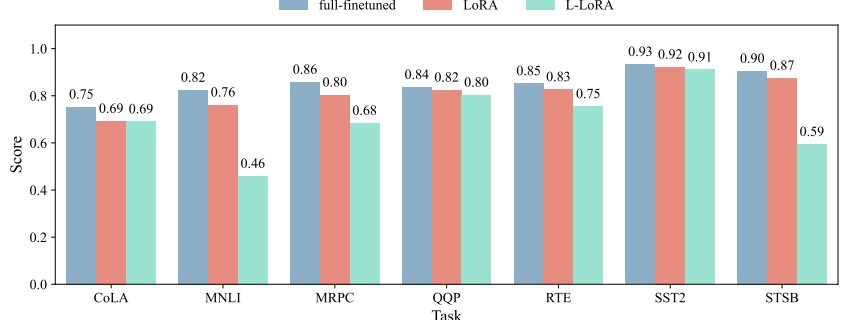

(b) Performance of task-specific Flan-T5-Base models on downstream NLP task.

Figure 7: Performance of full fine-tuning, LoRA, and L-LoRA on downstream tasks. Full fine-tuning achieves the best individual task performance, followed by LoRA and then L-LoRA which trades off some task-specific accuracy for improved parameter efficiency.

# D    LOSS LANDSCAPE AND COS SIMILARITY BETWEEN TASK VECTORS

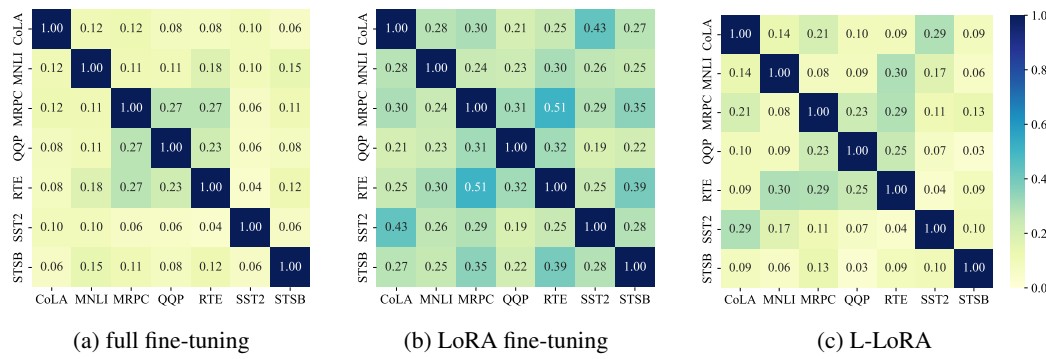

(a) full fine-tuning                    (b) LoRA fine-tuning                    (c) L-LoRA

Figure 8: **Similarity heatmap of task vectors**. These figures show heatmaps of the cosine similarity between task vectors from task-specific Flan-T5 model (Chung et al., 2022) fine-tuned on different NLP tasks from GLUE benchmark (Wang et al., 2018). The task vector represents the parameter differences between the fine-tuned models and the original pre-trained model. (a) Cos similarity matrix of task vectors when using full fine-tuning of the entire model. (b) Task vector similarities when using LoRA, a parameter-efficient fine-tuning method. (c) Cos similarity of task vectors from L-LoRA, our proposed linearized tuning approach that fine-tunes parameter-efficient modules in tangent space.

Figures 2 and 8 visualize the similarity between task-specific models fine-tuned on different tasks using full fine-tuning, LoRA, and linearized LoRA (L-LoRA). Full fine-tuning results in a low similarity between task vectors, indicating greater independence between task-specific representations. In contrast, LoRA tuning induces higher similarity and entanglement between tasks. L-LoRA improves over LoRA by further reducing the cosine similarity, demonstrating that linearized tuning of parameter-efficient modules enhances disentanglement compared to standard LoRA. The results are consistent across both a vision model (CLIP) fine-tuned on image datasets, and a language model (Flan-T5) fine-tuned on NLP datasets. This supports the hypothesis that constrained optimization techniques like L-LoRA can improve representational disentanglement compared to unstructured tuning approaches.

## E  MULTI-TASK MODEL FUSION

### E.1  FURTHER DETAILS ABOUT BASELINES

This section provides further implementation and configuration details for the baseline methods been compared in our experiments. An overview of the four baseline methods is provided as following. In Appendixes E.1.1, E.1.2, E.1.3 and E.1.4, the parameter settings used for each method in the experiments are explained.

- **Simple Weight Average**: We average the trainable weights of the models fine-tuned on different tasks with different fine-tune methods, this method is also referred to as ModelSoup (Wortsman et al., 2022a) and AdapterSoup (Chronopoulou et al., 2023) in the literature. The averaged model is then evaluated on the validation set of each task. For full fine-tuned models, the weights of the models are averaged directly (i.e., let $\theta = 1/n \sum_{i=1}^{n} \theta_i$). For parameter-efficient fine-tuning, we average the added weights and then apply to the parameter-efficient modules, i.e., let $\phi = \frac{1}{n} \sum_{i=1}^{n} \phi_i$ and $f(x) = f_{\theta_0}(x; \phi)$.
- **Task Arithmetic** (Ilharco et al., 2023; Zhang et al., 2023): As defined in Section 3.1, the task vector is computed on the set of trainable parameters. We compute the task vector for each task and then add them to construct a multi-task vector. The multi-task vector is then multiplied by a scaling factor $\lambda$ element-wisely and added to the initial parameters of the pre-trained model to obtain a multi-task model, i.e. $\theta = \theta_0 + \lambda \sum_i (\theta_i - \theta_0)$ for full fine-tuning and $\phi = \phi_0 + \lambda \sum_i (\phi_i - \phi_0), f(x) = f_{\theta_0}(x; \phi)$ for parameter-efficient fine-tuning, where $\lambda$ is a hyperparameter that the best-performing model is chosen on validation set.
- **Ties-Merging** (Yadav et al., 2023): Ties-merging is a method for merging multiple task-specific models into a single multi-task model. This algorithm follows three steps (trim, elect sign of parameters, and disjoint merge) to obtain a merged task vector $\nu$. Given the final merged task vector $\nu$, the final model is chosen in a similar way as task arithmetic, i.e. $\theta = \theta_0 + \lambda \nu$ for full fine-tuning and $\phi = \phi_0 + \lambda \nu, f(x) = f_{\theta_0}(x; \phi)$ for parameter-efficient fine-tuning, where $\lambda$ is a hyperparameter that the best-performing model is chosen on validation set.
- **LoraHub** (Huang et al., 2023): Lorahub is a method to apply arithmetic operations to parameter-efficient modules. With the help of a few examples from a novel task, LoraHub estimates the optimal weights $\{w_i\}_i$ for the parameter-efficient modules by utilizing a black-box optimization technique (Liu et al., 2020; Sun et al., 2022) to minimize $\mathcal{L} + \alpha \sum_i |w_i|$, where $\alpha$ is a hyperparameter. The final model is $f(x) = f_{\theta_0}(x; \phi)$, where $\phi = \sum_i w_i \phi_i \ \ s.t. \sum_i \alpha_i = 1$. In practical implementation, we construct the final model with task vector arithmetic because we have $\nu = \phi - \phi_0 = (\sum_i w_i \phi_i) - \phi_0 = \sum_i w_i \nu_i$, so $\phi = \phi_0 + \sum_i w_i \nu_i$. We evaluate LoraHub with both LoRA fine-tuning and L-LoRA fine-tuning.

### E.1.1  SIMPLE WEIGHT AVERAGE

In our comparison, we employ the method known as ModelSoup (Wortsman et al., 2022a) or AdapterSoup (Chronopoulou et al., 2023) in the literature, which involves averaging the trainable weights of models. We adapt simple averaging to models that have been fine-tuned on different tasks using different fine-tuning methods. By averaging the weights in this manner, we create a combined model that incorporates the knowledge and expertise gained from multiple tasks and training methods. This averaged model is subsequently evaluated on the validation set of each respective task to assess its performance.

For full fine-tuned models, we directly average the weights by computing the mean, represented as $\theta = \frac{1}{n}\sum_{i=1}^{n}\theta_i$, where $n$ is the total number of models being averaged.

When it comes to parameter-efficient fine-tuning, we compute the average of the added weights and then apply them to the parameter-efficient modules. This is accomplished by calculating the mean of the added weights as $\phi = \frac{1}{n}\sum_{i=1}^{n}\phi_i$, and utilizing this average weight configuration in the parameter-efficient modules. Consequently, the function $f(x)$ is defined as $f(x) = f_{\theta_0}(x; \phi)$, taking into account the updated averaged weights. We create a total of $2^7 - 8 = 120$ multi-task models for each fine-tuning method.

### E.1.2 TASK ARITHMETIC

As defined in Section 3.1, the task vector is computed on the set of trainable parameters. We compute the task vector for each task and then add them to construct a multi-task vector. The multi-task vector is then multiplied by a scaling factor $\lambda$ element-wisely and added to the initial parameters of the pre-trained model to obtain a multi-task model, i.e. $\theta = \theta_0 + \lambda \sum_i (\theta_i - \theta_0)$ for full fine-tuning (Ilharco et al., 2023) and $\phi = \phi_0 + \lambda \sum_i (\phi_i - \phi_0), f(x) = f_{\theta_0}(x; \phi)$ for parameter-efficient fine-tuning (Zhang et al., 2023), where $\lambda$ is a hyperparameter that the best-performing model is chosen on validation set.

Following the practice from (Ilharco et al., 2023), we add task vectors together and use a single scaling coefficient for the sum of vectors, $\lambda \in \{0, 0.05, 0.1, \ldots, 1.0\}$. That is, for each set of tasks from the power set of $\{\tau_i\}_{i=1}^{7}$, we construct 21 candidate models with different $\lambda$ values. We then evaluate the performance of each model on the validation set of each combination of tasks, and choose the best-performing model for each combination of tasks. Finally, we evaluate the performance of the best-performing model on the test set of each task. Here we have created a total of $21 \times (2^7 - 8) = 2520$ multi-task models for each fine-tuning method.

### E.1.3 TIES-MERGING

The Ties-Merging method, proposed in the paper by Yadav *et al.* (Yadav et al., 2023), is employed for merging multiple task-specific models into a unified multi-task model. This merging process involves three key steps: trim, elect sign of parameters, and disjoint merge, ultimately resulting in a merged task vector denoted as $\nu$. Once the final merged task vector $\nu$ is obtained, the selection of the final model parallels the approach used in task arithmetic. For full fine-tuning, the updated parameters are computed as $\theta = \theta_0 + \lambda\nu$, while for parameter-efficient fine-tuning, the updated parameters are computed as $\phi = \phi_0 + \lambda\nu$, and the function $f(x)$ is defined as $f_{\theta_0}(x; \phi)$. As previously mentioned, the hyperparameter $\lambda$ is chosen based on the performance of the model on the validation set.

We sweep the hyperparameters $k$ from the parameter trim step and scaling factor over a range of values $\{0.25, 0.50, 0.75, 1.00\}$. This results in a total of 16 combinations, whereby each combination is evaluated to determine the best-performing model. Here we have created a total of $4^2 \times (2^7 - 8) = 1920$ multi-task models for each fine-tune method.

### E.1.4 LORAHUB

LoraHub, as introduced by Huang *et al.* (Huang et al., 2023), is a method specifically designed for applying arithmetic operations to parameter-efficient modules. This technique leverages a set of example inputs from a novel task to estimate the optimal weights $\{w_i\}_i$ for the parameter-efficient modules. To achieve this, LoraHub utilizes a black-box optimization technique, such as the one described in the works of Liu et al. (Liu et al., 2020) and Sun *et al.* (Sun et al., 2022). The optimization process aims to minimize the objective function $\mathcal{L} + \alpha \sum_i |w_i|$, where $\alpha$ is a hyperparameter.

The final model obtained through LoraHub is defined as $f(x) = f_{\theta_0}(x; \phi)$, where $\phi = \sum_i w_i \phi_i$ subject to the constraint $\sum_i \alpha_i = 1$. In practical implementations, the final model is constructed using task vector arithmetic. This is possible due to the relationship $\nu = \phi - \phi_0 = (\sum_i w_i \phi_i) - \phi_0 = \sum_i w_i \nu_i$, which allows us to express $\phi$ as $\phi = \phi_0 + \sum_i w_i \nu_i$.

We adapt the original implementation form (Huang et al., 2023) and fix the hyperparameter $\alpha$ at 0.05. The maximum inference step for the gradient-free optimization algorithm Shiwa is 40, which results in up to $40 \times (2^7 - 8) = 4800$ multi-task models for each fine-tune method.

Table 5: Normalized scores of seven-task model fusion (CLIP-ViT-B/16) across individual datasets.

| Method | Cars | DTD | EuroSAT | GTSRB | RESISC45 | SUN397 | SVHN | Mean |
|--------|------|-----|---------|-------|----------|--------|------|------|
| | | | *Simple Average* | | | | | |
| FFT | 0.77 | 0.62 | **0.76** | 0.49 | 0.71 | 0.87 | **0.56** | **0.68** |
| LoRA | 0.81 | 0.69 | 0.39 | 0.40 | 0.62 | 0.93 | 0.16 | 0.57 |
| L-LoRA | **0.87** | **0.86** | 0.41 | **0.58** | **0.74** | **0.97** | 0.19 | 0.66 |
| | | | *Task Arithmetic* | | | | | |
| FFT | 0.80 | 0.63 | **0.87** | 0.73 | 0.78 | 0.86 | **0.85** | 0.79 |
| LoRA | 0.86 | 0.72 | 0.51 | 0.50 | 0.73 | **0.91** | 0.48 | 0.67 |
| L-LoRA | **0.95** | **0.91** | 0.65 | **0.74** | **0.86** | 0.96 | 0.78 | **0.84** |
| | | | *Ties-Merging* | | | | | |
| FFT | 0.78 | 0.65 | **0.92** | 0.69 | **0.83** | 0.88 | **0.93** | **0.81** |
| LoRA | 0.81 | 0.74 | 0.54 | 0.50 | 0.71 | 0.92 | 0.40 | 0.66 |
| L-LoRA | **0.89** | **0.91** | 0.60 | 0.69 | 0.82 | **0.97** | 0.51 | 0.77 |
| | | | *LoraHub* | | | | | |
| FFT | 0.93 | **0.90** | 0.32 | 0.30 | 0.47 | 0.77 | 0.13 | 0.55 |
| LoRA | 0.83 | 0.68 | 0.38 | 0.76 | 0.70 | 0.86 | 0.64 | 0.69 |
| L-LoRA | **0.94** | 0.88 | **0.73** | **0.85** | **0.85** | **0.91** | **0.80** | **0.85** |

### E.2 EXPERIMENTAL RESULTS VISUALIZATION AND DISCUSSIONS

We fine-tune all models with hyperparameter settings as shown in Table 2. We construct multi-task models by utilizing task vectors specific to individual tasks, employing various model fusion algorithms. Due to the extensive volume of data and multitude of models in the validation set of downstream NLP tasks, feasibility constraints led us to run only the first 50 batches for each NLP task. This approach allowed us to manage computational resources while still obtaining a representative evaluation within practical limits.

In Figures 10 and 11, we examine four different model fusion techniques - simple averaging, task arithmetic, ties merging, and LoraHub - for constructing multi-task models using an increasing number of task vectors on both vision and language domains.

Simple averaging doesn't account for the relationships or interactions between different tasks. As more tasks are combined, destructive interference may occur between incompatible task vectors. This indicates that simply averaging task vectors is insufficient for effectively combining knowledge across many NLP tasks. In contrast, the other three fusion methods demonstrate steady gains or maintain average multi-task model performance as the number of task vectors increases.

For image classification tasks, ties-merging exhibits the strongest overall performance, with the highest average normalized scores across different numbers of task vectors. This may be because ties-merging introduces several innovations to address the challenges of parameter interference during model merging. The higher degree of interference between vision tasks, evident in the cosine similarity matrix in Figures 2 and 8, increases the importance of these techniques.

In contrast, for NLP tasks, LoraHub demonstrates the strongest performance with the highest average normalized scores. The learned routing between shared fixed parameters and task-specific adapters enables more effective fusion for language tasks.

In Table 5, we present the normalized scores of seven-task model fusion (CLIP-ViT-B/16) across individual datasets. We observe that L-LoRA outperforms LoRA and FFT in most cases. This is because L-LoRA fine-tuning is able to better disentangle the task-specific knowledge from the shared knowledge. Besides, different fusion techniques can be more aligned with certain parameter-efficient methods, leading to substantial performance divergence depending on the chosen approach.

Another key observation from Figures 11 is that LoRA performs better than L-LoRA when the number of task vectors being fused is small. As the number of task vectors increases, the average

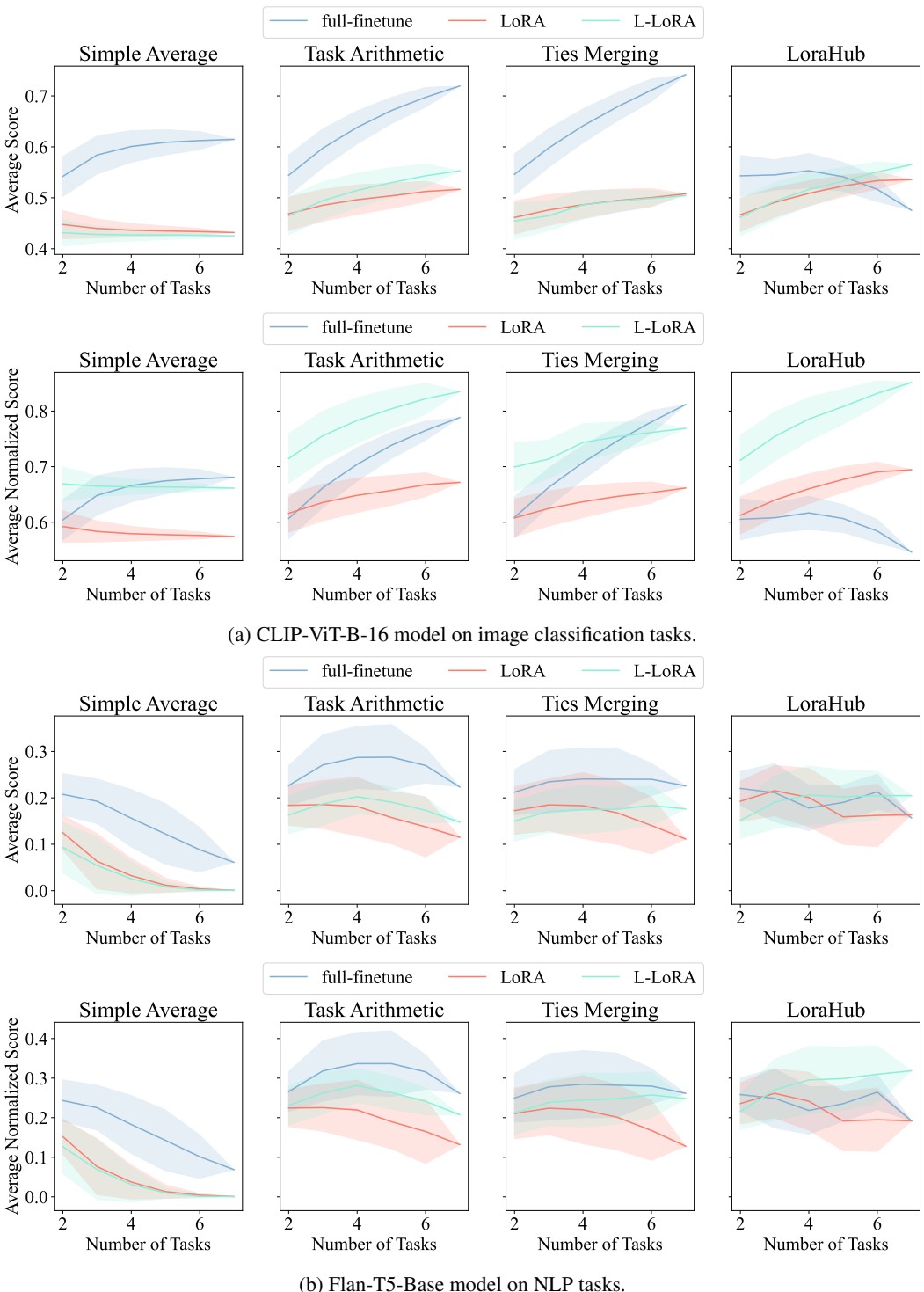

Figure 9: **Multi-task model fusion**: we construct multi-task models by utilizing task vectors specific to individual tasks, employing various model fusion algorithms. In the evaluation, the x-axis represents the number of task vectors used in building the multi-task model, while the y-axis represents the average (normalized) scores of the multi-task models across all seven downstream tasks. The line on the plot represents the average (normalized) scores of all multi-task models when considering a fixed number of tasks, while the shaded area corresponds to the standard deviation. The y-axis ticks are shared across these plots. A more detailed comparison between LoRA and L-LoRA is shown in Figures 10 and 11.

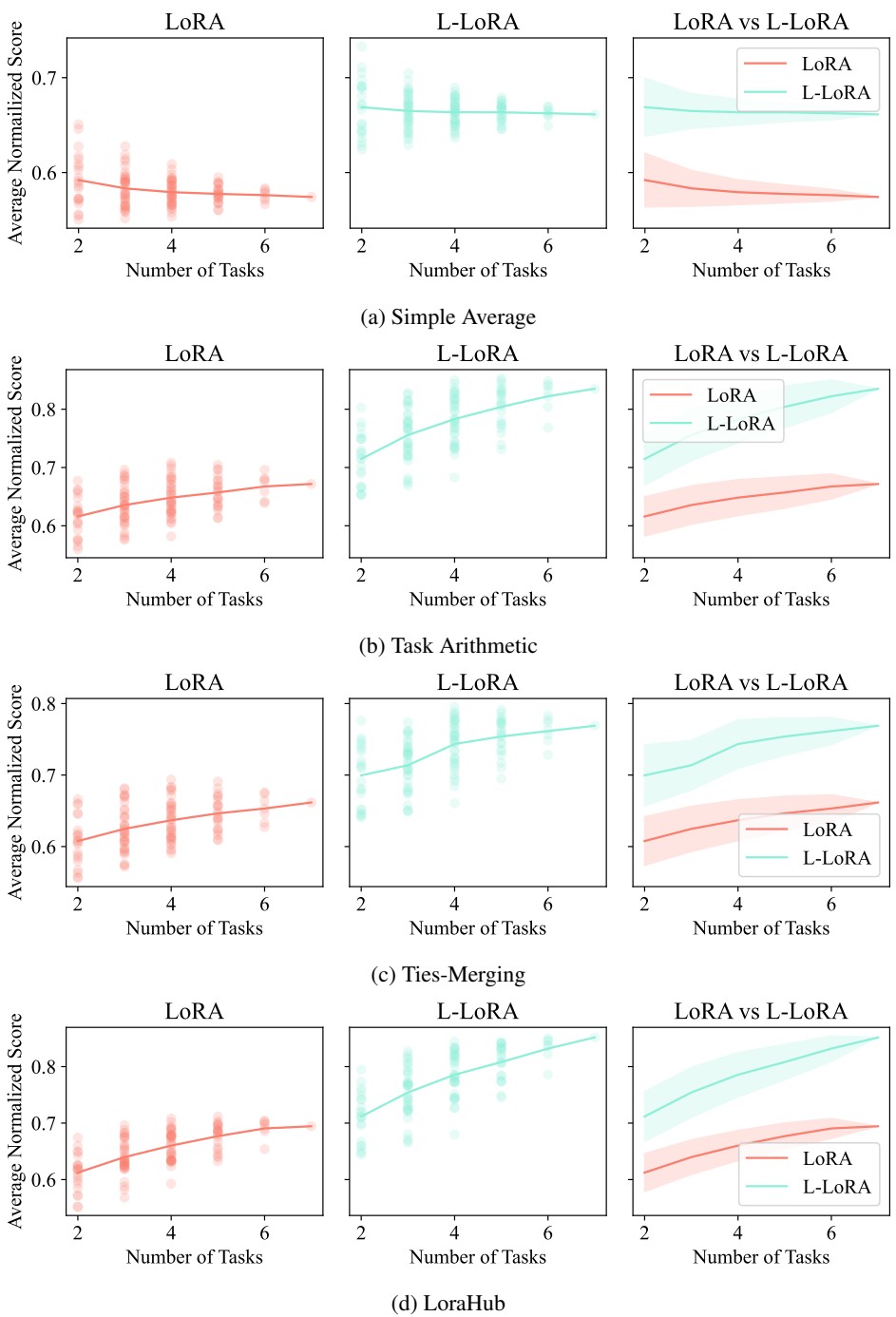

Figure 10: **LoRA vs L-LoRA (CLIP-ViT-B-16 model on vision tasks).** We construct multi-task models by utilizing task vectors specific to individual tasks, employing various model fusion algorithms. In the evaluation, the x-axis represents the number of task vectors used in building the multi-task model, while the y-axis represents the average score of the multi-task model across the seven GLUE tasks. Each point on the plot corresponds to the average score of a specific multi-task model, and the line represents the average score of all multi-task models considering a fixed number of tasks.

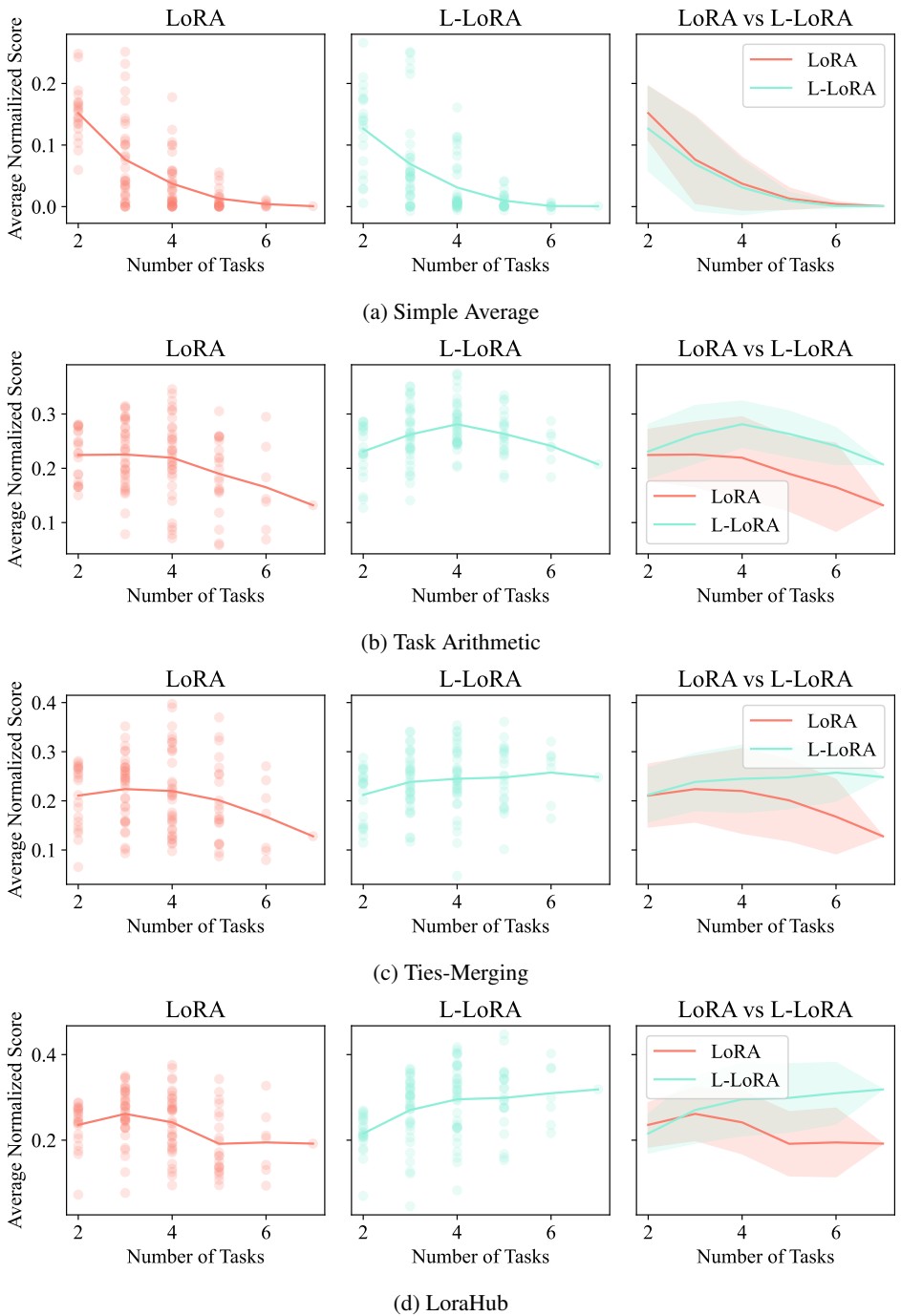

Figure 11: **LoRA vs L-LoRA (Flan-T5-Base model on NLP tasks)**. We construct multi-task models by utilizing task vectors specific to individual tasks, employing various model fusion algorithms. In the evaluation, the x-axis represents the number of task vectors used in building the multi-task model, while the y-axis represents the average score of the multi-task model across the seven GLUE tasks. Each point on the plot corresponds to the average score of a specific multi-task model, and the line represents the average score of all multi-task models considering a fixed number of tasks.

performance of L-LoRA becomes better than that of LoRA. It becomes evident that LoRA excels in fusing a smaller number of tasks or leveraging its effectiveness through simple averaging. On the other hand, L-LoRA shows an enhanced ability to exploit interactions within a larger set of tasks. This is because the linearization of PEFT modules in L-LoRA is fine-tuned in tangent space, which is a linear space, while the PEFT modules in LoRA are fine-tuned in the original non-linear ambient space. The non-linear space fine-tuning results in a superior performance of the single-task model, which is also known as the non-linear advantage (Guillermo Ortiz-Jimenez et al., 2023). However, as the number of tasks increases, L-LoRA becomes more effective than LoRA. This suggests that the learned linear combination of task vectors used in L-LoRA is able to leverage interactions between a larger number of tasks more effectively, and L-LoRA is better able to scale to handle multi-task model fusion across a large number of tasks.

## F    IMPLEMENTATION DETAILS

Here, we provide more details about the implementation of our method, including some basic code.

### F.1    PREPROCESSED EXAMPLES

In this subsection, we provide some examples of the preprocessed data for the downstream tasks. We use the same preprocessing steps as described in (Raffel et al., 2020).

#### F.1.1    COLA

We assign label 0 as "*unacceptable*" and label 1 as "*acceptable*".

Original inputs:

- **sentence:** Our friends won't buy this analysis, let alone the next one we propose.
- **label:** 1

Preprocessed:

- **input:** *cola sentence:* Our friends won't buy this analysis, let alone the next one we propose.
- **target:** *acceptable*

#### F.1.2    MNLI

We assign label 0 as "*entailment*", label 1 as "*neutral*" and label 2 as "*contradiction*".

Original inputs:

- **hypothesis:** Product and geography are what make cream skimming work.
- **premise:** Conceptually cream skimming has two basic dimensions - product and geography.
- **label:** 1

Preprocessed:

- **input:** *mnli hypothesis:* Product and geography are what make cream skimming work. *premise:* Conceptually cream skimming has two basic dimensions - product and geography.
- **target:** *neutral*

#### F.1.3    MRPC

We assign label 0 as "*not_quivalent*" and label 1 as "*equivalent*".

Original inputs:

- **sentence1:** Amrozi accused his brother , whom he called "the witness" , of deliberately distorting his evidence .
- **sentence2:** Referring to him as only "the witness" , Amrozi accused his brother of deliberately distorting his evidence .
- **label:** 1

Preprocessed:

- **input:** *mrpc sentence1:* Amrozi accused his brother , whom he called "the witness" , of deliberately distorting his evidence . *sentence2:* Referring to him as only "the witness" , Amrozi accused his brother of deliberately distorting his evidence .
- **output:** *equivalent*

### F.1.4 QQP

We assign label 0 as "*not_duplicate*" and label 1 as "*duplicate*".

Original inputs:

- **question1:** How is the life of a math student? Could you describe your own experiences?
- **question2:** Which level of prepration is enough for the exam jlpt5?
- **label:** 0

Preprocessed:

- **input:** *qqp question1:* How is the life of a math student? Could you describe your own experiences? *question2:* Which level of prepration is enough for the exam jlpt5?
- **output:** *not_duplicate*

### F.1.5 RTE

We assign label 0 as "entailment" and label 1 as "not_entailment".

Original inputs:

- **sentence1:** No Weapons of Mass Destruction Found in Iraq Yet.
- **sentence2:** Weapons of Mass Destruction Found in Iraq.
- **label:** 1

Preprocessed:

- **input:** *rte sentence1:* No Weapons of Mass Destruction Found in Iraq Yet. *sentence2:* Weapons of Mass Destruction Found in Iraq.
- **output:** *not_entailment*

### F.1.6 SST2

We assign label 0 as "*negative*" and label 1 as "*positive*".

Original inputs:

- **sentence:** hide new secretions from the parental units
- **label:** 0

Preprocessed:

- **input:** *sst2 sentence:* hide new secretions from the parental units
- **output:** *negative*

### F.1.7 STSB

We format the 'label' argument as a string with one decimal place, the corresponding Python code is x'"{:.1f}".format(label)'.

Original inputs:

- **sentence1:** A plane is taking off.
- **sentence2:** An air plane is taking off.
- **label:** 5

Preprocessed:

- **input:** *stsb sentence1:* A plane is taking off. *sentence2:* An air plane is taking off.
- **output:** *5.0*

### F.2 DISCUSSION ON THE CHOICE OF PROMPT TEMPLATES

Table 6: Comparison of the templates used in our experiments and a better version of prompt templates.

| Task | Prompt Templates | input text | target text |
|------|------------------|------------|-------------|
| **CoLA** | Templates in Appendix F.1 | "cola sentence: {sentence}" | "unacceptable" if label=0 else "acceptable" |
| | Improved Prompt Templates | "Indicate if the following sentence is grammatically correct or not: "{sentence}". **Answere 'acceptable' or 'unacceptable'.**" | "unacceptable" if label=0 else "acceptable" |
| **MNLI** | Templates in Appendix F.1 | "mnli hypothesis: {hypothesis} premise: {premise}" | "entailment", "neutral", "contradiction" if label is 0, 1,2, respectively |
| | Improved Prompt Templates | "Does the premise: '{premise}' logically imply, contradict, or is neutral to the hypothesis: '{hypothesis}'? **Answer with 'entailment', 'contradiction', or 'neutral'.**" | "entailment", "neutral", "contradiction" if label is 0, 1,2, respectively |
| **RTE** | Templates in Appendix F.1 | "rte sentence1: {sentence1} sentence2: {sentence2}" | "entailment" if label=0 else "not_entailment" |
| | Improved Prompt Templates | "Does the text: '{sentence1}' entail that '{sentence2}' is true? **Provide 'yes' or 'no'.**" | "yes" if label=0, else "no" |

To optimize the model's output, providing more explicit cues within the input is essential. An illustrative prompt, such as "sentence. Is this sentence 'positive' or 'negative'?" as suggested in Appendix F.1.6, signals clearly to the language model that the desired response should be categorized as either 'positive' or 'negative', potentially leading to improved accuracy.

To validate this approach, we conducted a set of focused multi-task model fusion experiments targeting three downstream tasks: CoLA, MNLI, and RTE. Across all fine-tuning processes, we employed the Adam optimizer, standardized the batch size to 16, and set the number of fine-tuning steps to 2000 for all models. While the learning rate was configured to 1e-5 for full fine-tuning, for both LoRA and L-LoRA, we increased the learning rate to 4e-5.

Table 6 provides a comparison between the original prompt templates and an enhanced version. Table 7 outlines the individual performance of fine-tuned models using different prompt templates.

Table 7: individual performance of fine-tuned models with different prompt templates

| Fine-tuning Method | CoLA | MNLI | RTE | Mean |
|---|---|---|---|---|
| *Prompt Templates in Appendix F.1* | | | | |
| full fine-tuning | 0.75 | 0.82 | 0.85 | 0.81 |
| LoRA fine-tuning | 0.69 | 0.76 | 0.83 | 0.76 |
| L-LoRA fine-tuning | 0.69 | 0.46 | 0.75 | 0.63 |
| *Better Prompt Templates* | | | | |
| full fine-tuning | 0.75 | 0.83(+1%) | 0.86(+1%) | 0.81 |
| LoRA fine-tuning | 0.69 | 0.83(+9%) | 0.84(+11%) | 0.79(+4%) |
| L-LoRA fine-tuning | 0.69 | 0.82(+78%) | 0.81(+29%) | 0.77(+22%) |

Table 8: Three-model fusion using simple model averaging, we evaluate 'exact-match' accuracy on validation datasets. These results show that a better and more predictable prompt yields superior results, with L-LoRA outperforming LoRA in terms of both absolute scores and normalized scores.

| | Prompt Templates in Appendix F.1 | | | | Improved Prompt Templates | | | |
|---|---|---|---|---|---|---|---|---|
| Method | CoLA | MNLI | RTE | Mean | CoLA | MNLI | RTE | MEAN |
| *Absolute Score* | | | | | | | | |
| FFT | **0.67** | **0.37** | **0.54** | **0.53** | **0.70** | **0.78** | **0.82** | **0.76** |
| LoRA | 0.35 | 0.00 | **0.42** | **0.25** | 0.69 | 0.63 | 0.82 | 0.71 |
| L-LoRA | 0.35 | 0.00 | 0.23 | 0.20 | 0.69 | **0.73** | 0.81 | **0.74** |
| *Normalized Score* | | | | | | | | |
| FFT | **0.90** | **0.45** | **0.63** | **0.66** | 0.93 | **0.94** | 0.95 | 0.94 |
| LoRA | 0.50 | 0.00 | **0.50** | **0.34** | 1.00 | 0.76 | 0.98 | 0.91 |
| L-LoRA | **0.51** | 0.00 | 0.31 | 0.28 | 1.00 | **0.89** | **1.00** | **0.96** |

In Table 8, we evaluate 'exact-match' accuracy on validation datasets through a simple model averaging technique for three-model fusion. Our results indicate that a more effective and predictable prompt yields superior performance, with L-LoRA outperforming LoRA in terms of both absolute scores and normalized scores. In the "Absolute Score" section, the full fine-tuning (FFT) method appears to show the highest scores with both sets of prompt templates, indicating superior performance in both original and improved settings. However, L-LoRA shows an increase in performance with the improved prompts compared to the original prompts, particularly in the MNLI and RTE columns and subsequently in the MEAN column. The "Normalized Score" section provides a relative comparison of each method's performance. In this section, it suggests that despite FFT showing high performance in absolute terms, L-LoRA demonstrates increased normalized scores with better prompt templates, indicating greater effectiveness on weight disentanglement when adjusted to more appropriate prompting.

This experiment demonstrats that the choice of prompt templates can impact the performance of these model fusion methods. It illustrates how a more effective prompting strategy, especially one that is better aligned with the model and task at hand, can lead to significant improvements in performance. Besides, the highest normalized scores with these better prompt templates are achieved by L-LoRA, suggesting that it is the most responsive to prompt optimization among the methods tested.

## F.3    Model Linearization

```python
class LinearizedModelWraper(nn.Module):
    def __init__(self, model: nn.Module, init_model: nn.Module = None):
        """
        Initializes a linearized model.

        Args:
            model (nn.Module): The underlying PyTorch model to be linearized.
            init_model (nn.Module): The initial PyTorch model used to compute the linearization parameters (
        default: None).
        """
        super().__init__()
        self.model = model
        if init_model is None:
            init_model = model
        assert not hasattr(self, "params0")
        params0 = deepcopy([(k, v.detach()) for k, v in init_model.named_parameters()])
        self.params0_keys = [k for k, v in params0]
        self.params0_values = nn.ParameterList([v for k, v in params0])
        for p in self.params0_values:
            p.requires_grad_(False)

    def tuple_params_to_dict(self, tuple_params):
        """
        Converts a tuple of parameters to a dictionary with keys corresponding to the parameter names.

        Args:
            tuple_params (Tuple[Tensor, ...]): A tuple of parameters.

        Returns:
            Dict[str, Tensor]: A dictionary with keys corresponding to the parameter names and values
        corresponding to the
            parameter values.
        """
        assert len(tuple_params) == len(self.params0_keys)
        state_dict = {}
        for k, p in zip(self.params0_keys, tuple_params):
            state_dict[k] = p
        return state_dict

    def forward(self, *args, **kwargs):
        """
        Computes the linearized model output using a first-order Taylor decomposition.

        Args:
            *args: Positional arguments to be passed to the model.
            **kwargs: Keyword arguments to be passed to the model.

        Returns:
            torch.Tensor: The output of the linearized model, computed using a first-order Taylor decomposition.
        """
        params0 = tuple(self.params0_values)
        params = dict_params_to_tuple(OrderedDict(self.named_parameters()))
        dparams = tuple(p - p0 for p, p0 in zip(params, params0))
        out, dp = jvp(
            lambda *param: functional_call(
                self.model, self.tuple_params_to_dict(param), args, kwargs
            ),
            params0,
            dparams,
        )
        return out + dp
```

Listing 1: Pytorch code to linearize a model.

The `LinearizedModelWraper` class linearizes a given PyTorch model using a first-order Taylor expansion. The linearized model is computed using the `forward` method, which takes positional and keyword arguments and returns the output of the linearized model. The `tuple_params_to_dict` method converts a tuple of parameters to a dictionary with keys corresponding to the parameter names.

With the help of `LinearizedModelWraper` class, we can linearize the LoRA modules in a pre-trained language model, as the following code snippet shows:

```python
def _get_submodules(model: nn.Module, key):
    """
    Retrieves the parent module, target module, and target module name for a given key in a PyTorch model.

    Args:
        model (nn.Module): The PyTorch model to retrieve submodules from.
        key (str): The key representing the submodule to retrieve.

    Returns:
        Tuple[nn.Module, nn.Module, str]: A tuple containing the parent module, target module, and target module
            name.
    """
```

```
12    parent = model.get_submodule(".".join(key.split(".")[:-1]))
13    target_name = key.split(".")[-1]
14    target = model.get_submodule(key)
15    return parent, target, target_name
16
17
18  def linearize_lora_model(model: nn.Module):
19      """
20      Linearizes the LoraLayer modules in a PyTorch model.
21
22      Args:
23          model (nn.Module): The PyTorch model to be linearized.
24
25      Returns:
26          nn.Module: The linearized PyTorch model.
27      """
28      for key, module in model.named_modules():
29          if isinstance(module, LoraLayer) and isinstance(module, nn.Linear):
30              log.debug(f"convert {key} to linearized lora layer")
31              parent, target, target_name = _get_submodules(model, key)
32              setattr(parent, target_name, LinearizedModelWraper(target))
33      return model
```

Listing 2: Linearize all the LoRA modules in a LLM

Where the _get_submodules function takes a PyTorch model and a key representing a submodule in the model and returns a tuple containing the parent module, the target module, and the target module name. The parent module is the module that contains the target module.

The linearize_lora_model function takes a PyTorch model and linearizes all LoraLayer modules in the model. If a LoraLayer module is also an instance of nn.Linear, it is converted to a LinearizedModelWraper object. The function returns the linearized PyTorch model.

