# OpenReview forum: "Parameter-Efficient Multi-Task Model Fusion with Partial Linearization"
_ICLR.cc/2024/Conference — ICLR 2024 poster_

### Official Review · Reviewer_vkjS · 2023-10-24

**Soundness:** 3 good
**Presentation:** 3 good
**Contribution:** 2 fair
**Rating:** 8
**Confidence:** 2

**Summary:**

This paper proposes a novel approach to enhance multi-task fusion in large pre-trained models. The authors introduce partial linearization of adapter modules combined with task arithmetic, improving the fusion of multiple tasks while maintaining efficient fine-tuning and inference. Experimental results demonstrate that this method outperforms standard techniques, especially as the number of tasks increases. The contribution lies in its ability to construct unified multi-task models effectively and efficiently fuse fine-tuned task vectors, highlighting the benefits of partial linearization for scalable multi-task model fusion.

**Strengths:**

- The paper exhibits a clear and logical structure, making it easy to comprehend.

- The proposal's effectiveness is demonstrated through comprehensive experiments conducted on both NLP and image classification tasks. The visualization provided in Figure 6 offers intriguing insights into disentanglement error, further substantiating the proposal's efficacy.

**Weaknesses:**

- The absence of experiments conducted on larger-scale models diminishes the significance of the proposal.

**Questions:**

- Can the proposed methods demonstrate effectiveness on SOTA models?

---

> ### Author Response · Authors · 2023-11-14
> **Response to reviewer vkjS**
>
> Thank you for your comprehensive review and the acceptance recommendation for our paper. Your feedback is greatly appreciated and will help us improve the final version of our paper.
>
> **W1: The absence of experiments conducted on larger-scale models diminishes the significance of the proposal.**
>
> **The scale of the experiments.** Regarding the scale of the experiments, we acknowledge that including larger-scale models could further validate the robustness and scalability of our proposal. While our current results already demonstrate the effectiveness of our method on CLIP and Flan-T5 models, we are eager to extend our experiments to SOTA models to provide a more thorough validation of our approach. Due to resource limitations, conducting experiments on larger models was challenging as they require substantial GPU resources, which were difficult to support. We futher report results of extra small-scale experiments of semantic segmentation task in response to Q1, which indicate our method can be extended to larger model and pixel-level vision task.
>
> ---
>
> **Q1: Can the proposed methods demonstrate effectiveness on SOTA models?**
>
> **Effectiveness on SOTA models.** We are confident that our partial linearization method can be applied to SOTA models to enhance their multi-task learning capabilities. The positive results we observed in CLIP and Flan-T5 models, which are themselves highly competitive, give us a strong indication that our method would be beneficial when applied to other advanced models. We will take your question as a valuable suggestion for future work and aim to include such results in subsequent publications.
>
> To demonstrate this, we conducted a small-scale segmentation experiment. We utilized the SAM model ('sam_vit_b_01ec64.pth') to test the segmentation tasks with both LoRA and L-LoRA on the Pascal VOC 2012 and NYUD v2 datasets. We set the batch size to 4 and the LoRA r hyperparameter to 32. *Due to time constraints and resource limitations, the models were still far from convergence at this point, but the results were sufficient to demonstrate the effectiveness of L-LoRA (The code for this experiment will be put on GitHub together with a clean version of the original code).* We report the performance of the pre-trained models, the fine-tuned models, and the fused models obtained through weight averaging on the validation sets of Pascal VOC and NYUD v2 in terms of mIoU in the table below.
>
> Table: reported mIoU of LoRA and L-LoRA.
>
> | Model                             | fine-tuning method | VOC 2012 | NYUD v2 | Mean mIoU |
> |-----------------------------------|--------------------|----------|---------|-----------|
> | Pretrained Model                  |                    | 0.65     | 0.75    | 0.70      |
> | fine-tuned (far from converge)    | LoRA               | **4.51**     | **2.42**    | **3.47**      |
> |                                   | L-LoRA             | 4.09     | 1.93    | 3.01      |
> | two-model fusion (simple average) | LoRA               | 1.92(43%)     | 1.15(47%)    | 1.53      |
> |                                   | L-LoRA             | **4.05(99%)**     | **1.51(78%)**    | **2.78**      |
>
> Notably, the two-model fusion using simple averaging yields different results for each method. The LoRA fusion only retains 43% and 47% of the pretrained model's performance on VOC 2012 and NYUD v2, respectively. In contrast, the L-LoRA fusion maintains a remarkable 99% and 78% of the fine-tuned LoRA's performance on the respective datasets, resulting in a mean mIoU of 2.78.

---

> > ### Comment · Reviewer_vkjS · 2023-11-22
> > **I will keep my rating**
> >
> > I would like to thank the authors for their rebuttal efforts to make the paper more solid. I will suggest the author add the results to the paper. My concerns are addressed and I will keep my rating of the paper.

---

> > > ### Author Response · Authors · 2023-11-23
> > >
> > > Thank you once again for your thoughtful evaluation and support.

---

### Official Review · Reviewer_HkFi · 2023-10-27

**Soundness:** 2 fair
**Presentation:** 3 good
**Contribution:** 2 fair
**Rating:** 6
**Confidence:** 4

**Summary:**

This paper proposes a novel method to improve multi-task fusion for parameter-efficient fine-tuning techniques like LoRA fine-tuning. Specifically, their approach partially linearizes only the adapter modules and applies task arithmetic over the linearized adapters. This allows us to leverage the advantages of model fusion over linearized fine-tuning, while still performing fine-tuning and inference efficiently. Extensive experiments are conducted.

**Strengths:**

1. The code is provided.
2. This paper proposes a new linearized LoRA method.
3. Extensive experiments are conducted.

**Weaknesses:**

1. The novelty of this paper is limited. This paper simply adapts the proposed method in [1] by replacing full fine-tuning with LoRA. Thus, the method in this paper is obviously more efficient than [1] since LoRA is more efficient than full fine-tuning.
2. The definition of the task vector of LoRA in this paper seems to be unreasonable.
3. The L-LoRA has a large performance drop on single-task fine-tuning (Figure 7) and a slight increase on the merged case (Table 1, especially in the NLP domain) compared to LoRA. Thus, it is unclear what are the advantages of L-LoRA compared to LoRA.

**Questions:**

### Major Concerns:
1. The novelty of this paper is limited. [1] studies the linearized full fine-tuning while this paper simply replaces full fine-tuning with LoRA. Although the authors emphasize the proposed method is more efficient than [1], it is obvious since LoRA is more efficient than fully fine-tuning.
2. What's the meaning of $\phi_0$? For the full fine-tuning, $\theta_0$ is the pre-trained model weight and is shared for every task $\tau_i$. However, for LoRA, $\phi_i$ is newly added parameters for each task $\tau_i$. Thus, what is $\psi_0$? Is $\psi_0$ a shared initialization for LoRA matrixes of every task $\tau_i$?
3. In [2], the task vector of full fine-tuning is defined as $\nu_i=\theta_i-\theta_0$, which means the parameter change of $\theta_0$. Thus, why the task vector of LoRA in this paper is defined as the change of LoRA parameters $\phi_i-\phi_0$ rather than the change of $\theta_0$ as in [3], i.e., $A_iB_i$, where $\phi_i=[A_i, B_i]$ is the LoRA parameters.
4. Table 1 only shows the average normalized scores over multiple datasets, so how about the performance of each dataset? Could you provide it in the Appendix?
5. It seems the proposed L-LoRA method does not perform well on the NLP domain, according to the results in Table 1 and the similarity heatmap in Figure 8.
6. Why use the proposed L-LoRA rather than the existing LoRA? The single-task fine-tuning results in Figure 7 show that L-LoRA has a large performance drop compared to LoRA in many datasets.

### Minor Concerns:
1. The caption of Figure 3(c): not linearized?


**References**

[1] Task Arithmetic in the Tangent Space: Improved Editing of Pre-Trained Models. arXiv:2305.12827.

[2] Editing Models with Task Arithmetic. ICLR, 2023.

[3] Effective and Parameter-Efficient Reusing Fine-Tuned Models. arXiv:2310.01886.

---

> ### Author Response · Authors · 2023-11-14
> **Response to reviewer HkFi (1)**
>
> Thank you for your detailed evaluation of our manuscript and for raising several insightful points. We appreciate the opportunity to address your concerns and clarify aspects of our work.
>
> **W1 & Q1: The novelty of this paper is limited. This paper simply adapts the proposed method in [1] by replacing full fine-tuning with LoRA. Thus, the method in this paper is obviously more efficient than [1] since LoRA is more efficient than full fine-tuning.**
>
> **Novelty of the paper.** We understand your concern regarding the perceived incremental nature of our contribution. However, we must emphasize that the essence of our contribution extends beyond mere efficiency improvements. Our work's innovation lies in the application of first-order information during the partial linearization of LoRA modules. This use of first-order information is pivotal; it captures the most critical aspects of model variability, which is the fundamental reason our method is effective, not merely an incremental extension.
>
> In our approach, we carefully considered the specific aspects of the LoRA fine-tuned model that would benefit most from linearization. Rather than applying a blanket linearization across the entire model, we strategically targeted the adapter modules for this process. This focused application of linearization to the adapters is a deliberate choice, driven by the understanding that these modules play a crucial role in task-specific adaptations within the larger model architecture. By linearizing only these components, we aim to enhance their ability to integrate and retain task-specific information without incurring the computational costs associated with linearizing the entire model. This method allows us to maintain the overall efficiency of the LoRA framework while reaping the benefits of linearization where it counts the most—within the adapters that are key to the model's fine-tuning for distinct tasks.
>
> [1] Task Arithmetic in the Tangent Space: Improved Editing of Pre-Trained Models. arXiv:2305.12827.
>
> ---
>
> **Q2: What's the meaning of $\phi_0$? For the full fine-tuning, $\theta_0$ is the pre-trained model weight and is shared for every task $\tau_i$. However, for LoRA, $\phi_i$ is newly added parameters for each task $\tau_i$. Thus, what is $\phi_0$? Is $\phi_0$ a shared initialization for LoRA matrixes of every task $\tau_i$?**
>
> **Meaning of $\phi_0$.** Yes, $\phi_0$ represents a shared initialization for LoRA matrixes across all tasks in our work. As detailed in Appendix B, we have employed a consistent random seed—specifically, seed 42—for initializing the parameter-efficient models in both the vision and language domains.
>
> The rationale behind this specific condition is that when defining the task vector for PEFT models, it is necessary to have a common starting point. In practice, when using only LoraHub to perform model fusion, it is possible to relax this constraint. This is because LoraHub computes a weighted sum of the parameters across the entire LoRA modules to produce a merged LoRA module $\phi = \sum_i w_i \phi_i$, thus avoiding the introduction of the concept of task vector.

---

> ### Author Response · Authors · 2023-11-14
> **Response to reviewer HkFi (2)**
>
> **W2 & Q3: The definition of the task vector of LoRA in this paper seems to be unreasonable. 1. In [2], the task vector of full fine-tuning is defined as $\nu_i=\theta_i - \theta_0$, which means the parameter change of $\theta_0$. Thus, why the task vector of LoRA in this paper is defined as the change of LoRA parameters $\phi_i-\phi_0$ rather than the change of $\theta_0$ as in [3], i.e., $A_i B_i$, where $\phi_i=[A_i, B_i]$ is the LoRA parameters.**
>
> **Task vector definition.** Our definition of the task vector as $\phi_i - \phi_0$ (the trainable parameter space, LoRA parameter space) rather than the change of $\theta$ (the original parameter space) as done in [3, 4] is a deliberate and calculated decision that aligns with the nature of our fine-tuning approach.
>
> 1. When partially linearizing a LoRA model, we concentrate on the dynamics of the trainable parameters and compute the Jacobian-verctor product for all trainable parameters, specifically the LoRA parameters denoted by $\phi$. The original model parameters, denoted by  $\theta_0$, are treated as constants during fine-tuning—akin to model buffers that provide a stable foundation upon which the adaptable components of the model, the LoRA parameters, can exert their influence.
> 2. From a mathematical standpoint, we perform a first-order Taylor expansion on the trainable parameters, which does not involve the parameter merging operations of the LoRA module.
>
>    $$f^{\text{lin}}\_{\theta_0}(x;\phi)=f\_{\theta_0}(x; \phi_0) + \nabla_\phi f\_{\theta_0}(x;\phi_0)^\top (\phi -\phi_0)$$
>
>    It's important to note that this process of Taylor expansion is applied exclusively to the parameters that are designated as trainable—those that we actively adjust during the fine-tuning phase.
> 3. Although our experiments have solely utilized LoRA fine-tuning, such a definition is helpful for generalize our method to other parameter-efficient fine-tuning techniques, such as adapter-tuning.
>
> [2] Editing Models with Task Arithmetic. ICLR, 2023.
> [3] Effective and Parameter-Efficient Reusing Fine-Tuned Models. arXiv:2310.01886.
> [4] Composing Parameter-Efficient Modules with Arithmetic Operations. http://arxiv.org/abs/2306.14870.
>
>
> ---
>
> **Q4: Table 1 only shows the average normalized scores over multiple datasets, so how about the performance of each dataset? Could you provide it in the Appendix?**
>
> **Performance on Each dataset.** We agree that a detailed breakdown of performance across individual datasets would be beneficial. We will include these results in the Appendix to provide a more comprehensive view of our method's performance.
>
> Table: normalized scores of merged CLIP-ViT-B/16 across individual datasets.
>
> |     Method      | Fine-tuning Mode |   Cars   |   DTD    | EuroSAT  |  GTSRB   | RESISC45 |  SUN397  |   SVHN   |   Mean   |
> | :-------------: | :--------------: | :------: | :------: | :------: | :------: | :------: | :------: | :------: | :------: |
> | Simple Average  | full fine-tuning |   0.77   |   0.62   | **0.76** |   0.49   |   0.71   |   0.87   | **0.56** | **0.68** |
> |                 |       LoRA       |   0.81   |   0.69   |   0.39   |   0.40   |   0.62   |   0.93   |   0.16   |   0.57   |
> |                 |      L-LoRA      | **0**.87 | **0.86** |   0.41   | **0.58** | **0.74** | **0.97** |   0.19   |   0.66   |
> | Task Arithmetic | full fine-tuning |   0.80   |   0.63   | **0.87** |   0.73   |   0.78   |   0.86   | **0.85** |   0.79   |
> |                 |       LoRA       |   0.86   |   0.72   |   0.51   |   0.50   |   0.73   | **0.91** |   0.48   |   0.67   |
> |                 |      L-LoRA      | **0.95** | **0.91** |   0.65   | **0.74** | **0.86** |   0.96   |   0.78   | **0.84** |
> |  Ties-Merging   | full fine-tuning |   0.78   |   0.65   | **0.92** |   0.69   | **0.83** |   0.88   | **0.93** | **0.81** |
> |                 |       LoRA       |   0.81   |   0.74   |   0.54   |   0.50   |   0.71   |   0.92   |   0.40   |   0.66   |
> |                 |      L-LoRA      | **0.89** | **0.91** |   0.60   |   0.69   |   0.82   | **0.97** |   0.51   |   0.77   |
> |     LoraHub     | full fine-tuning |   0.93   | **0.90** |   0.32   |   0.30   |   0.47   |   0.77   |   0.13   |   0.55   |
> |                 |       LoRA       |   0.83   |   0.68   |   0.38   |   0.76   |   0.70   |   0.86   |   0.64   |   0.69   |
> |                 |      L-LoRA      | **0.94** |   0.88   | **0.73** | **0.85** | **0.85** | **0.91** | **0.80** | **0.85** |

---

> ### Author Response · Authors · 2023-11-14
> **Response to reviewer HkFi (3)**
>
> **Q5: It seems the proposed L-LoRA method does not perform well on the NLP domain, according to the results in Table 1 and the similarity heatmap in Figure 8.**
>
> **Performance in the NLP Domain.** The observed underperformance in the NLP domain could be due to our initial choice of prompt templates not being optimal for fine-tuning the language models—a slight oversight in our experimental design. Nevertheless, we are confidence that this does not skew the comparison of multi-task model fusion performance between various methods, given that the same prompt template was employed throughout the experiments.
>
> To enhance the model's output, we should provide more explicit cues within the input. For instance, as suggested in Appendix E.1.6, a more instructive prompt would be "{sentence}. Is this sentence 'positive' or 'negative'?" This type of prompt would clearly signal to the language model that the response should be either 'positive' or 'negative', potentially resulting in increased accuracy.
>
> To demonstrate this, we carried out a series of small-scale multi-task model fusion experiments focusing on three downstream tasks: CoLA, MNLI, and RTE. For all fine-tuning processes, we utilized the Adam optimizer, standardizing the batch size to 16 and setting the number of fine-tuning steps to 2000 for all models. The learning rate was configured to 1e-5 for full fine-tuning, while for both LoRA and L-LoRA, we increased the learning rate to 4e-5. Table 1 presents a comparison between the prompt templates originally used in our study and an improved version of these templates. Table 2 details the individual performance of fine-tuned models employing different prompt templates. In Table 3, we assess the 'exact-match' accuracy on validation datasets through a simple model averaging technique for three-model fusion. Our findings indicate that a better and more predictable prompt yields superior results, with L-LoRA outperforming LoRA in terms of both absolute scores and normalized scores.
>
> Table 1: Comparison of the templates used in our paper and a better version of prompt templates.
>
> | Task | Prompt Templates        | input text                                                                                                                                                                      | target text                                                                 |
> | ---- | ----------------------- | ------------------------------------------------------------------------------------------------------------------------------------------------------------------------------- | --------------------------------------------------------------------------- |
> | CoLA | Templates in Appendix E | "cola sentence: {sentence}"                                                                                                                                                     | "unacceptable" if label=0 else "acceptable"                                 |
> |      | Better Prompt Templates | "Indicate if the following sentence is grammatically correct or not:   \"{sentence}\". **Answere 'acceptable' or 'unacceptable'.**"                                             | "unacceptable" if label=0 else "acceptable"                                 |
> | MNLI | Templates in Appendix E | "mnli hypothesis: {hypothesis} premise: {premise}"                                                                                                                              | "entailment", "neutral", "contradiction" if   label is 0, 1,2, respectively |
> |      | Better Prompt Templates | "Does the premise: '{premise}' logically imply, contradict, or is   neutral to the hypothesis: '{hypothesis}'? **Answere with 'entailment',   'contradiction', or 'neutral'.**" | "entailment", "neutral", "contradiction" if   label is 0, 1,2, respectively |
> | RTE  | Templates in Appendix E | "rte sentence1: {sentence1} sentence2: {sentence2}"                                                                                                                             | "entailment" if label=0 else "not_entailment"                               |
> |      | Better Prompt Templates | "Does the text: '{sentence1}' entail that '{sentence2}' is true? **Provide   'yes' or 'no'.**"                                                                                  | "yes" if label=0, else "no"                                                 |

---

> ### Author Response · Authors · 2023-11-14
> **Table 2 and Table 3 of 'Response to reviewer HkFi (3)'**
>
> Table 2: individual performance of fine-tuned models with different prompt templates
>
> |                                | Fine-tuning Method | CoLA | MNLI       | RTE        | Mean       |
> | ------------------------------ | ------------------ | ---- | ---------- | ---------- | ---------- |
> | Prompt Templates in Appendix E | full fine-tuning   | 0.75 | 0.82       | 0.85       | 0.81       |
> |                                | LoRA fine-tuning   | 0.69 | 0.76       | 0.83       | 0.76       |
> |                                | L-LoRA fine-tuning | 0.69 | 0.46       | 0.75       | 0.63       |
> | Better Prompt Templates        | full fine-tuning   | 0.75 | 0.83(+1%)  | 0.86(+1%)  | 0.81       |
> |                                | LoRA fine-tuning   | 0.69 | 0.83(+9%)  | 0.84(+11%) | 0.79(+4%)  |
> |                                | L-LoRA fine-tuning | 0.69 | 0.82(+78%) | 0.81(+29%) | 0.77(+22%) |
>
> Table 3: three-model fusion using simple model averaging, we evaluate 'exact-match' accuracy on validation datasets.  These results show that a better and more predictable prompt yields superior results, with L-LoRA outperforming LoRA in terms of both absolute scores and normalized scores. (v0: prompt templates in Appendix E; v1: better prompt templates.)
>
> |                  |                    | v0 |          |          |          |    | v1 |          |          |          |
> |------------------|--------------------|--------------------------------|----------|----------|----------|----|-------------------------|----------|----------|----------|
> |                  | fine-tuning method | CoLA                           | MNLI     | RTE      | Mean     | \| | CoLA                    | MNLI     | RTE      | MEAN     |
> | Absolute Score   | full-finetuning    | **0.67**                       | **0.37** | **0.54** | **0.53** | \| | **0.70**                | **0.78** | **0.82** | **0.76** |
> |                  | LoRA               | 0.35                           | 0.00     | **0.42** | **0.25** | \| | 0.69                    | 0.63     | 0.82     | 0.71     |
> |                  | L-LoRA             | 0.35                           | 0.00     | 0.23     | 0.20     | \| | 0.69                    | **0.73** | 0.81     | **0.74** |
> | Normalized Score | full-finetuning    | **0.90**                       | **0.45** | **0.63** | **0.66** | \| | 0.93                    | **0.94** | 0.95     | 0.94     |
> |                  | LoRA               | 0.50                           | 0.00     | **0.50** | **0.34** | \| | 1.00                    | 0.76     | 0.98     | 0.91     |
> |                  | L-LoRA             | **0.51**                       | 0.00     | 0.31     | 0.28     | \| | 1.00                    | **0.89** | **1.00** | **0.96** |
>
>
> Additionally, the performance discrepancies might also stem from inherent differences in task characteristics and dataset nature within the NLP field.

---

> ### Author Response · Authors · 2023-11-14
> **Response to reviewer HkFi (4)**
>
> **W3 & Q6: The L-LoRA has a large performance drop on single-task fine-tuning (Figure 7) and a slight increase on the merged case (Table 1, especially in the NLP domain) compared to LoRA. Thus, it is unclear what are the advantages of L-LoRA compared to LoRA. Why use the proposed L-LoRA rather than the existing LoRA?**
>
> **Advantages of L-LoRA over LoRA.** Despite the performance drop in single-task fine-tuning observed in Figure 7, L-LoRA demonstrates improved performance in the multi-task fusion setting. The increase in performance in the merged case, especially in the NLP domain, suggests that L-LoRA may offer advantages in scenarios where multi-task performance is prioritized.
>
> The multi-task domain is particularly challenging due to the need to balance and optimize across various objectives. The performance gains achieved by L-LoRA in this setting suggest that our method can effectively navigate the trade-offs between tasks, finding what could be described as a Pareto optimal point—a solution where no task's performance can be improved without compromising another's. The advantages of L-LoRA become evident when considering the broader context of its application. The advantages of L-LoRA become evident when considering the broader context of its application. While single-task performance is important, the ability to excel in multi-task environments is increasingly becoming a benchmark for success in advanced machine learning systems. Our method contributes to this goal by enhancing multi-task model fusion capabilities, which is a significant step forward in developing more capable and efficient AI systems.
>
> By the way, based on the small-scale model fusion experiments we conducted in response to Q5, we observe that superior prompt templates can significantly narrow the performance gap between L-LoRA and LoRA. But searching for better prompt templates is not the focus of our research in this paper.
>
> ---
>
> **Minor concern 1: caption of figure 3(c)**
>
> This is a typo, thanks for pointing it out and we will correct it. "(c) Linearized..." should be "(d) Linearized..."

---

> > ### Comment · Reviewer_HkFi · 2023-11-22
> >
> > Thanks to the authors for the rebuttal. I still have some concerns about my initial comments.
> >
> > 1. Novelty: If my understanding is correct, this paper uses the method in [1] in LoRA fine-tuning instead of the full fine-tuning in [1]. Are there any technical challenges when adapting the method in [1] to LoRA?
> >
> > 2. Task vector of LoRA: Since the definition of the task vector for LoRA in this paper is different from other works like [3, 4], I suggest the authors discuss it in the paper. And $\phi_0$ should be clearly defined in the paper.
> >
> > 3. Since LoRA matrixes across all tasks share the same initialization $\phi_0$, do you conduct repeated experiments with different $\phi_0$ or explore the effect of $\phi_0$?
> >
> > 3. The feedback of Q4: (1) why the mean results in this table are different from the one in Table 1 of the paper? (2) why do the results of L-LoRA/LoRA have a large difference with full fine-tuning in different datasets? For example, using the simple average method, the results of L-LoRA in DTD and SVHN are 0.86 and 0.19, respectively, but the results of full fine-tuning are 0.62 and 0.56. (3) have you added this table to the revised paper?
> >
> > 4. The NLP experiments: I suggest the authors add the results with new prompt templates to the paper.

---

> > > ### Author Response · Authors · 2023-11-22
> > > **Response to Reviewer HkFi (1)**
> > >
> > > Thank you for your thoughtful review and constructive feedback on our paper. As per your suggestions, our paper has undergone significant improvement!
> > >
> > > **Q1: Novelty: If my understanding is correct, this paper uses the method in [1] in LoRA fine-tuning instead of the full fine-tuning in [1]. Are there any technical challenges when adapting the method in [1] to LoRA?**
> > >
> > > We selectively identify and apply linearization to specific parameters, enhancing efficiency and enabling theoretical analysis. Besides, adopting partial linearization resulting in significantly improved speed and reduced GPU memory usage than full linearization.
> > >
> > > 1. **Identify linearization parts**. In LoRA, only a sparse, low-rank structure of the weight matrices is updated. We need to identify which aspects of the model’s parameters will benefit from linearization while full fine-tuning methods have more freedom as all parameters are fine-tuned. In our experiments, LoRA adapt to the attention weights ($W=W_0+AB$). Our linearization did not extend to the entire attention weights $W$ but was specifically applied to the LoRA parameters $\phi=[A,B]$. This not only makes linearization more efficient but also facilitates theoretical analysis. We have the following:
> > >    $$\phi(\Delta t)-\phi(0) \propto \mathbb{E}\_{(x_{\tau_i},y_{\tau_i})\sim D\_{\tau_i}} [
> > > \boldsymbol{K}(x, x_{\tau_i}; \phi(0))
> > > \nabla_f \mathcal{L}\_{\text{CE}}(f\_{\theta_0}(x\_{\tau_i}; \phi(0)), y\_{\tau_i})].$$
> > > 	This formula underscores the direct correspondence between the changes in LoRA weights $[A,B]$ of a linearized model and the gradient of the loss. It hints at the underlying mechanisms of weight disentanglement, revealing that the adjustment in LoRA parameters is essentially directed by the aggregated gradient information from the task-specific data. This understanding is crucial as it provides insight into how a model's weights evolve in response to the learning process.
> > >
> > > 1. **Computation overhead**. Implementing linearization across the entire model (linearized full fine-tuning) significantly increases memory usage and inference cost. Through partially linearization, we have significantly reduced the overhead associated with linearization. The training and inference speed of partial linearization is 2-3 times faster than that of full linearization, with GPU memory usage only being one-third to one-half of the latter.
> > >
> > > Table 1: Summary of training and inference cost.
> > >
> > > | **Scenario**                                          | **Training (Adam)** |      | **Inference**           |      |
> > > | ------------------------------------------------------ | ------------------- | ---- | ------------------------ | ---- |
> > > |                                                        | **Time**            | **VRAM** | **Time**           | **VRAM** |
> > > | *CLIP-ViT-B-32 (batch size=64)*                        |                     |      |                          |      |
> > > | full fine-tuning   | 8.25 it/s           | $\approx 4.2$GB | 13.28 it/s           | $\approx 0.9$GB |
> > > | full linearization | 4.94 it/s           | $\approx 6.3$GB | 8.52 it/s            | $\approx 2.0$GB |
> > > | LoRA (r=16)        | 8.43 it/s           | $\approx 2.6$GB | 13.01 it/s           | $\approx 0.9$GB |
> > > | L-LoRA (r=16)      | 7.55 it/s           | $\approx 2.7$GB | 10.65 it/s           | $\approx 1.0$GB |
> > > | *CLIP-ViT-B-16 (batch size=64)*                        |                     |      |                          |      |
> > > | full fine-tuning                                       | 2.97 it/s           | $\approx 12.4$GB | 8.92 it/s            | $\approx 1.6$GB |
> > > | full linearization                                     | 1.49 it/s           | $\approx 20.1$GB | 2.50 it/s            | $\approx 3.7$GB |
> > > | LoRA (r=16)                                            | 3.83 it/s           | $\approx 9.1$GB  | 8.60 it/s            | $\approx 1.6$GB |
> > > | L-LoRA (r=16)                                          | 3.24 it/s           | $\approx 9.2$GB  | 7.23 it/s            | $\approx 1.8$GB |
> > > | *CLIP-ViT-L-14 (batch size=16)*                        |                     |      |                          |      |
> > > | full fine-tuning                                       | 2.55 it/s           | $\approx 13.9$GB | 8.23 it/s            | $\approx 1.8$GB |
> > > | full linearization                                     | 1.32 it/s           | $\approx 21.8$GB | 2.26 it/s            | $\approx 4.8$GB |
> > > | LoRA (r=16)                                            | 3.49 it/s           | $\approx 9.1$GB  | 7.99 it/s            | $\approx 1.8$GB |
> > > | L-LoRA (r=16)                                          | 2.99 it/s           | $\approx 9.3$GB  | 6.55 it/s            | $\approx 1.9$GB |

---

> > > > ### Author Response · Authors · 2023-11-22
> > > > **Response to Reviewer HkFi (2)**
> > > >
> > > > **Q2: Task vector of LoRA: Since the definition of the task vector for LoRA in this paper is different from other works like [3, 4], I suggest the authors discuss it in the paper. And $\phi_0$ should be clearly defined in the paper.**
> > > >
> > > > We add section to discuss the definition of task vector for PEFT models to Appendix A.
> > > > We appreciate your comment on the definition of the task vector for LoRA and its contrast with other works [3, 4]. We recognize the importance of clarifying the differences in our approach and intend to address this in our paper with a more detailed discussion.
> > > >
> > > > ---
> > > >
> > > > **Q3: Since LoRA matrixes across all tasks share the same initialization $\phi_0$, do you conduct repeated experiments with different $\phi_0$ or explore the effect of $\phi_0$?**
> > > >
> > > > In our current study, the initializations of LoRA matrices across all tasks, denoted by $\phi_0$, were kept consistent to ensure a controlled environment for comparing the performance of our proposed method. We used a fixed random seed 42 for this initialization, which is a common practice in machine learning experiments to enable reproducibility of results.
> > > >
> > > > ---
> > > >
> > > > **Q4: 1. The feedback of Q4:**
> > > > **(1) why the mean results in this table are different from the one in Table 1 of the paper?**
> > > >
> > > > In Table 1 of the paper, we computed the average results across all downstream tasks for all task combinations, amounting to a total of $2^7 - (7+1)$ multi-task combinations. However, in the Q4 response, due to the limitations of a single table in showcasing numerous results, we only presented the average value for one specific combination, namely the seven-task model fusion. It is worth noting that the average results in the Q4 response appear to be better than those in Table 1 of the paper. This discrepancy is attributed to the fact that when a task is missing, the performance of the merged model tends to be slightly inferior on the omitted task, contributing to the observed difference in average performance.
> > > >
> > > > **(2) why do the results of L-LoRA/LoRA have a large difference with full fine-tuning in different datasets? For example, using the simple average method, the results of L-LoRA in DTD and SVHN are 0.86 and 0.19, respectively, but the results of full fine-tuning are 0.62 and 0.56.**
> > > >
> > > > Both the complexity of tasks and the size of the training dataset are crucial factors. Due to the substantial difference in the number of parameters between LoRA (L-LoRA) and full fine-tuning, the effectiveness of full fine-tuning tends to surpass that of parameter-efficient fine-tuning methods when there is a large amount of data available for fine-tuning. The training set for DTD consists of a total of 3760 samples, while SVHN utilizes 73257 samples, significantly surpassing the training set size of DTD.
> > > >
> > > > In addition to the characteristics of datasets, different fusion techniques can be more aligned with certain fine-tuning methods, leading to substantial performance divergence depending on the chosen approach. Taking LoraHub as an example, because LoraHub employs a gradient-free method to determine the weight coefficients of fine-tuned models, its performance significantly drop when optimizing in high-dimensional spaces. As the model's parameter count increases, the dimensionality of the optimization problem also rises, leading to the full fine-tuning model fusion performing worse than LoRA.
> > > >
> > > > **(3) have you added this table to the revised paper?**
> > > >
> > > > We add this table along with a discussion paragraph to Appendix E.2.
> > > >
> > > > **Q5: The NLP experiments: I suggest the authors add the results with new prompt templates to the paper.**
> > > >
> > > > We add the results with new prompt templates to the appendix F.2.
> > > >
> > > > ---
> > > >
> > > > ***Thanks again for your feedback. We hope this clarification can address your concern.***

---

> > > > > ### Comment · Reviewer_HkFi · 2023-11-22
> > > > >
> > > > > Thanks to the authors for further feedback, which addresses most of my concerns and enhances my understanding of this paper. I am willing to raise my score now.
> > > > >
> > > > > But I still would like to emphasize the novelty of this paper is limited: this paper adapts an existing method in [1] to the LoRA parameters rather than the original whole model parameters in [1] and thus, the efficiency advantage in this paper is apparent because the LoRA parameters are much fewer than the original model parameters.

---

> > > > > > ### Author Response · Authors · 2023-11-23
> > > > > >
> > > > > > Dear reviewer, thank you for your valuable comments and support for our work.

---

### Official Review · Reviewer_iJMj · 2023-11-01

**Soundness:** 3 good
**Presentation:** 4 excellent
**Contribution:** 2 fair
**Rating:** 6
**Confidence:** 2

**Summary:**

This paper proposes a new method for Parameter-Efficient Fine Tuning (PEFT) of large pre-trained foundational models for multi-task models. The authors build on prior work in weight disentanglement (Ortiz-Jimenez et al. 2023) and extend it to LoRA for better fusion of models. The authors hypothesise that partial linearisation of a model through the LoRA modules during fine-tuning can improve weight disentanglement, which is conducive to better task arithmetic. Results are shown on a variety of experiments whereupon the proposed models outperforms others on vision classification tasks.

**Strengths:**

1. This is a very well written paper, motivations are clear, results are (mostly) well presented and it is clear to understand the results.

2. The idea to perform partial linearisation on LoRA modules is interesting and nicely presented. The results, especially on CLIP-ViT-B-16 (Figure 5a) are compelling argument to the original hypothesis of the work being correct.

**Weaknesses:**

1. A key result of the paper is the result from Appendix A which allows the authors to hypothesize that partial linearization of a subset of module parameters (here LoRA) can improve weight disentanglement.

    1a. This result needs to be in the main body of the text and properly explained. Without, it is difficult to
    understand exactly why the authors claim this.

    1b. Having checked the derivation in Appendix A, the authors show that the model output of a linearised
    model is only determined by the gradient of the loss in the non-linearised model on task $t_i$. How
    exactly then does this allow the authors to make the central hypothesis, which guides the presented
    method?

2. The results on the NLP task (Flan T-5-Base) need to be better explain. The method (L-LoRA) not only performs worse than full fine-tuning (so does LoRA fine-tuning) but also worse on average than LoRA. Why is this? Is the presented model only applicable to vision tasks?

**Questions:**

1. Section 4.2 introduces parameter scaling laws for weight disentanglement. However, what does this have to do with the method or key results? The scaling laws suggest over-parameterisation is necessary for weight disentanglement. I struggle to see the connection between this and the need to partially linearize a model for fine-tuning.

2. A remark in the manuscript is made bottom of page 8 that "...higher cosine similarity...implies greater redundancy and overlap...This results in more destructive task interference with naïve merging". I don't follow this as it seems to be opposite to most work in multi-task training. In that setting, naïve training would favour similar tasks as it would not require methods to mitigate task interference (see GradNorm method). Why is this then opposite in the context of this paper?

---

> ### Author Response · Authors · 2023-11-14
> **Response to reviewer iJMj (1)**
>
> We are grateful for your insightful comments and the opportunity to clarify the aspects of our work that you found unclear. Your feedback is invaluable in helping us refine our paper. Thank you for your thorough review and for considering our work.
>
> ---
>
> **W1: A key result of the paper is the result from Appendix A which allows the authors to hypothesize that partial linearization of a subset of module parameters (here LoRA) can improve weight disentanglement.**
> 1. **W1a: This result needs to be in the main body of the text and properly explained. Without, it is difficult to understand exactly why the authors claim this.**
>
> **Key result placement.** We acknowledge your suggestion regarding the placement of the key result from Appendix A. We will provide a more detailed explanation in section 4.
>
> 2. **W1b: Having checked the derivation in Appendix A, the authors show that the model output of a linearized model is only determined by the gradient of the loss in the non-linearized model on task $\tau_i$. How exactly then does this allow the authors to make the central hypothesis, which guides the presented method?**
>
> **Connection between linearization and weight disentanglement.** We agree that the derivation is central to our hypothesis and warrants more visibility. We will clarify this explanation in the manuscript to make the connection more explicit.
>
> The derivation in Appendix A shows that the change of the linearized model output is determined by the gradient of the loss. The connection to central hypothesis is that this property of linearized models enables a more predictable and controlled update during fine-tuning. By fine-tuning in the linearized space, updates of trainable parameters are perpendicular to the loss contours, this orthogonality promotes that the updates contribute to the separation of the weight representations corresponding to distinct tasks, a concept known as weight disentanglement.
>
> Mathematically, the change of linearized model output for a input sample $x$ around the initialization $\phi(0)$ can be written as:
>
> $$
> f^{\text{lin}}_{\theta_0}(x; \theta(\Delta t))-f\_{\theta_0}(x;\theta(0))
> =\nabla\_{\theta_0}f(x;\theta_0)^\top (\theta(\Delta t)-\theta(0))
> =-\eta \mathbb{E}\_{(x\_{\tau_i},y\_{\tau_i})\sim D\_{\tau_i}} [
> \boldsymbol{K}(x, x\_{\tau_i}; \theta(0))
> \nabla_f \mathcal{L}\_{\text{CE}}(f\_{\theta_0}(x\_{\tau_i}; \theta(0)), y\_{\tau_i})]
> $$
>
> Thus we have the following formula that captures the relationship between task-specific gradient of the loss and weight disentanglement for linearized model
>
> $$\theta(\Delta t)-\theta(0) \propto \mathbb{E}\_{(x\_{\tau_i},y\_{\tau_i})\sim D\_{\tau_i}}
> [\boldsymbol{K}(x, x\_{\tau_i}; \theta(0))
> \nabla_f \mathcal{L}\_{\text{CE}}(f\_{\theta_0}(x_{\tau_i}; \theta(0)), y_{\tau_i})]$$
>
> ---

---

> > ### Comment · Reviewer_iJMj · 2023-11-23
> >
> > Thanks for your response. I’ve read the rebuttal across all reviews.
> >
> > In view of the rebuttal - I will change my score from 5 to 6

---

> > > ### Author Response · Authors · 2023-11-23
> > > **Response to Reviewer iJMj**
> > >
> > > Thank you once again for your time.

---

> ### Author Response · Authors · 2023-11-14
> **Response to reviewer iJMj (2)**
>
> **W2: The results on the NLP task (Flan T-5-Base) need to be better explain. The method (L-LoRA) not only performs worse than full fine-tuning (so does LoRA fine-tuning) but also worse on average than LoRA. Why is this? Is the presented model only applicable to vision tasks?**
>
> **Performance on NLP Tasks.** Our selection of prompts was rather naive, following the approach used by (Raffel et al. in 2020), which was a minor oversight in our experimental design. Nonetheless, we are confident that this does not compromise the comparative analysis of multi-task model fusion performance across different methods, as the same prompt template was consistently applied in all experiments. We find that with fine-tuning of the prompt selection and other task-specific adjustments, L-LoRA can be made effective in the NLP domain.
>
> To demonstrate this, we carried out a series of small-scale multi-task model fusion experiments focusing on three downstream tasks: CoLA, MNLI, and RTE. For all fine-tuning processes, we utilized the Adam optimizer, standardizing the batch size to 16 and setting the number of fine-tuning steps to 2000 for all models. The learning rate was configured to 1e-5 for full fine-tuning, while for both LoRA and L-LoRA, we increased the learning rate to 4e-5. Table 1 presents a comparison between the prompt templates originally used in our study and an improved version of these templates. Table 2 details the individual performance of fine-tuned models employing different prompt templates. In Table 3, we assess the 'exact-match' accuracy on validation datasets through a simple model averaging technique for three-model fusion. Our findings indicate that a better and more predictable prompt yields superior results, with L-LoRA outperforming LoRA in terms of both absolute scores and normalized scores.
>
> Table 1: Comparison of the templates used in our paper and a better version of prompt templates.
>
> | Task | Prompt Templates | input text | target text |
> | ---- | ----------------------- | ------------------------------------------------------------------------------------------------------------------------------------------------------------------------------- | --------------------------------------------------------------------------- |
> | CoLA | Templates in Appendix E | "cola sentence: {sentence}"                                                                                                                                                     | "unacceptable" if label=0 else "acceptable" |
> | | Better Prompt Templates | "Indicate if the following sentence is grammatically correct or not:   \"{sentence}\". **Answere 'acceptable' or 'unacceptable'.**" | "unacceptable" if label=0 else "acceptable" |
> | MNLI | Templates in Appendix E | "mnli hypothesis: {hypothesis} premise: {premise}"                                                                                                                              | "entailment", "neutral", "contradiction" if   label is 0, 1,2, respectively |
> | | Better Prompt Templates | "Does the premise: '{premise}' logically imply, contradict, or is   neutral to the hypothesis: '{hypothesis}'? **Answere with 'entailment',   'contradiction', or 'neutral'.**" | "entailment", "neutral", "contradiction" if   label is 0, 1,2, respectively |
> | RTE  | Templates in Appendix E | "rte sentence1: {sentence1} sentence2: {sentence2}"                                                                                                                             | "entailment" if label=0 else "not_entailment" |
> | | Better Prompt Templates | "Does the text: '{sentence1}' entail that '{sentence2}' is true? **Provide   'yes' or 'no'.**"                                                                                  | "yes" if label=0, else "no" |

---

> ### Author Response · Authors · 2023-11-14
> **Table 2 & Table 3 of 'Response to reviewer iJMj (2)'**
>
> Table 2: individual performance of fine-tuned models with different prompt templates
>
> | | Fine-tuning Method | CoLA | MNLI | RTE | Mean |
> | ------------------------------ | ------------------ | ---- | ---------- | ---------- | ---------- |
> | Prompt Templates in Appendix E | full fine-tuning   | 0.75 | 0.82       | 0.85       | 0.81|
> | | LoRA fine-tuning| 0.69 | 0.76| 0.83 | 0.76|
> | | L-LoRA fine-tuning | 0.69 | 0.46| 0.75 | 0.63|
> | Better Prompt Templates| full fine-tuning   | 0.75 | 0.83(+1%)  | 0.86(+1%)  | 0.81|
> || LoRA fine-tuning   | 0.69 | 0.83(+9%)  | 0.84(+11%) | 0.79(+4%)  |
> || L-LoRA fine-tuning | 0.69 | 0.82(+78%) | 0.81(+29%) | 0.77(+22%) |
>
> Table 3: three-model fusion using simple model averaging, we evaluate 'exact-match' accuracy on validation datasets. (v0: prompt templates in Appendix E; v1: better prompt templates.)
>
> |                  |                    | v0 |          |          |          |    | v1 |          |          |          |
> |------------------|--------------------|--------------------------------|----------|----------|----------|----|-------------------------|----------|----------|----------|
> |                  | fine-tuning method | CoLA                           | MNLI     | RTE      | Mean     | \| | CoLA                    | MNLI     | RTE      | MEAN     |
> | Absolute Score   | full-finetuning    | **0.67**                       | **0.37** | **0.54** | **0.53** | \| | **0.70**                | **0.78** | **0.82** | **0.76** |
> |                  | LoRA               | 0.35                           | 0.00     | **0.42** | **0.25** | \| | 0.69                    | 0.63     | 0.82     | 0.71     |
> |                  | L-LoRA             | 0.35                           | 0.00     | 0.23     | 0.20     | \| | 0.69                    | **0.73** | 0.81     | **0.74** |
> | Normalized Score | full-finetuning    | **0.90**                       | **0.45** | **0.63** | **0.66** | \| | 0.93                    | **0.94** | 0.95     | 0.94     |
> |                  | LoRA               | 0.50                           | 0.00     | **0.50** | **0.34** | \| | 1.00                    | 0.76     | 0.98     | 0.91     |
> |                  | L-LoRA             | **0.51**                       | 0.00     | 0.31     | 0.28     | \| | 1.00                    | **0.89** | **1.00** | **0.96** |

---

> ### Author Response · Authors · 2023-11-14
> **Response to reviewer iJMj (3)**
>
> **Q1: Section 4.2 introduces parameter scaling laws for weight disentanglement. However, what does this have to do with the method or key results? The scaling laws suggest over-parameterisation is necessary for weight disentanglement. I struggle to see the connection between this and the need to partially linearize a model for fine-tuning.**
>
> **Parameter Scaling Law.** The parameter scaling law is an empirical phenomenon suggesting that models with a larger number of trainable parameters tend to exhibit better performance in multi-task model fusion. Consequently, a logical deduction would be that parameter-efficient fine-tuning could potentially diminish the performance of multi-task model fusion. Indeed, our experimental results corroborate this (as illustrated in Figure 5).
>
> However, our method stands out in that the performance of L-LoRA's multi-task model fusion remains on par with that of full fine-tuning, despite its parameter efficiency. This indicates that partial linearization, with a very limited increase in inference cost overhead, effectively bridges the gap typically observed between parameter efficiency and multi-task fusion performance.
>
> ---
>
> **Q2: A remark in the manuscript is made bottom of page 8 that "...higher cosine similarity...implies greater redundancy and overlap...This results in more destructive task interference with naïve merging". I don't follow this as it seems to be opposite to most work in multi-task training. In that setting, naïve training would favour similar tasks as it would not require methods to mitigate task interference (see GradNorm method). Why is this then opposite in the context of this paper?**
>
> **cos similarity and task interference.** You've raised an insightful point. When we discuss the impact of high cosine similarity on model performance, it's important to distinguish between the contexts of multi-task training and model fusion. *In the former, tasks with similar characteristics often benefit from the ability to share representations, as this commonality can facilitate learning by leveraging the same features or patterns.*
>
> However, in the context of model fusion, which is the focus of our paper, the dynamics differ from simultaneous multi-task training. *Here, we are dealing with models that have been independently trained on separate tasks and are subsequently being merged.* In such a post-hoc fusion, a high cosine similarity between task weights is indicative of overlap in the information encoded by the models, thus be problematic. While this might suggest a common underlying structure, it also raises the risk that merging these similar weights could amplify shared errors or redundancies, rather than complementing each model's strengths. It can signal redundancy, where multiple models encode similar information without adding new insights, or even conflict, where the shared information is not entirely compatible across tasks.
>
> In summary, while shared representations are advantageous in simultaneous multi-task learning, our work suggests that for the fusion of independently trained models, distinctiveness in task representations—achieved through orthogonality—is key to preserving the integrity and performance of the fused model. This distinction is crucial for understanding the different implications of cosine similarity in multi-task training versus model fusion contexts.

---

### Official Review · Reviewer_tr5D · 2023-11-01

**Soundness:** 3 good
**Presentation:** 2 fair
**Contribution:** 3 good
**Rating:** 8
**Confidence:** 2

**Summary:**

Efficient finetuning on the pretrained large model has been an important topic. In this work, a partial linearization method (L-Lora) is proposed under the context of PERF(parameter-efficient finetuning).  The key idea is applying linearization to adapter modules and applies task arithmetic over the linearized adapters. In practice, first-order Tayler expansion is used to linearize the model dynamics at time $t$. Based on the derivation from a neural tangent kernel theory, the hypothesis is that partial linearization of a subset of model parameters during fine-tuning can also improve weight disentanglement compared to full non-linear fine-tuning. CLIP and Flan-T5 are used to verify the hypothesis in vision-language and language domains.

**Strengths:**

- Evaluations are conducted on both vision-language and language tasks.
- The proposed method achieved significant performance improvement, compared with the standard LoRA strategy.

**Weaknesses:**

- In vision domain, only the high-level vision task like image classification tasks evaluated, the mid-level and low level task are missing, for example, semantic segmentation.

**Questions:**

- From table 1, seems the L-LoRA method are outperforming full-finetuning under some model fusion settings, do we have some possible illustrations?
- Is it possible to evaluate L-LoRA on image segmentation foundational models like SAM?

---

> ### Author Response · Authors · 2023-11-14
> **Response to reviewer tr5D**
>
> Thank you for your thoughtful review and constructive feedback on our paper. We are grateful for your positive assessment of our work and appreciate the strengths you've highlighted.
>
> **Q1: From table 1, seems the L-LoRA method are outperforming full-finetuning under some model fusion settings, do we have some possible illustrations?**
>
> **Results from Table 1.** Indeed, as observed in Table 1, our L-LoRA (Linearized Low-Rank Adaptation) appears to outperform full fine-tuning in terms of normalized score under certain fusion settings.
>
> On one hand, the partially linearization likely leads to better weight disentanglement*, meaning that the task-specific knowledge encoded in the weights is more separable.
>
> On the other hand, we should consider *the data requirement of full fine-tuning versus parameter-efficient fine-tuning. As for fine-tuning, the amount of data in downstream tasks is sometimes not enough. However, full fine-tuning generally demands extensive data to optimize the large number of parameters effectively. In scenarios where abundant data is available, full fine-tuning is expected to excel as it can leverage the comprehensive dataset to refine a large amount of model's weights for each specific task. However, in situations with limited data, full fine-tuning may not perform optimally due to overfitting concerns or insufficient information for the model.
>
> L-LoRA, on the other hand, with its focus on updating only a subset of the model's parameters, is less dependent on large amounts of data. This characteristic could explain why L-LoRA sometimes outperforms full fine-tuning in multi-task fusion settings, especially when data availability of downstream dataset is a limiting factor.
>
> ---
>
> **W1 & Q2: In vision domain, only the high-level vision task like image classification tasks evaluated, the mid-level and low level task are missing, for example, semantic segmentation. Is it possible to evaluate L-LoRA on image segmentation foundational models like SAM?**
>
> **Extending evaluation on mid-level and low-level task (Evaluating on image segmentation models).** We acknowledge the limitation you pointed out regarding the scope of our evaluations in the vision domain. Our decision to focus on high-level vision tasks like image classification was driven by the desire to establish a strong foundational understanding of multi-task model fusion in a well-studied context, and we agree that extending the evaluation to lower-level vision tasks would provide a more comprehensive understanding.
>
> Evaluating our method on mid-level and low-level task, such as image segmentation, is indeed possible. This is due to the task-agnostic nature of our approach, which does not impose specific requirements on the loss function. The only prerequisite for our method is that the tuned parameter-efficient modules must be capable of computing the Jacobian-vector product, thereby enabling it to be efficiently linearized.
>
> To demonstrate this, we conducted a small-scale experiment. We utilized the SAM model ('sam_vit_b_01ec64.pth') to test the segmentation tasks with both LoRA and L-LoRA on the Pascal VOC 2012 and NYUD v2 datasets. We set the batch size to 4 and the LoRA r hyperparameter to 32. *Due to time constraints and resource limitations, the models were still far from convergence at this point, but the results were sufficient to demonstrate the effectiveness of L-LoRA (The code for this experiment will be put on GitHub together with a clean version of the original code).* We report the performance of the pre-trained models, the fine-tuned models, and the fused models obtained through weight averaging on the validation sets of Pascal VOC and NYUD v2 in terms of mIoU in the table below.
>
> Table: reported mIoU of LoRA and L-LoRA.
>
> | Model                             | fine-tuning method | VOC 2012 | NYUD v2 | Mean mIoU |
> |-----------------------------------|--------------------|----------|---------|-----------|
> | Pretrained Model                  |                    | 0.65     | 0.75    | 0.70      |
> | fine-tuned (far from converge)    | LoRA               | **4.51**     | **2.42**    | **3.47**      |
> |                                   | L-LoRA             | 4.09     | 1.93    | 3.01      |
> | two-model fusion (simple average) | LoRA               | 1.92(43%)     | 1.15(47%)    | 1.53      |
> |                                   | L-LoRA             | **4.05(99%)**     | **1.51(78%)**    | **2.78**      |
>
> Notably, the two-model fusion using simple averaging yields different results for each method. The LoRA fusion only retains 43% and 47% of the pretrained model's performance on VOC 2012 and NYUD v2, respectively. In contrast, the L-LoRA fusion maintains a remarkable 99% and 78% of the fine-tuned LoRA's performance on the respective datasets, resulting in a mean mIoU of 2.78.

---

> > ### Comment · Reviewer_tr5D · 2023-11-22
> > **response to rebuttal**
> >
> > Thanks for the detailed response and the new experiment results. I will keep my rating and it will be great if you can add these results to the paper

---

> > > ### Author Response · Authors · 2023-11-23
> > >
> > > Thank you for your suggestions to improve our paper and your final support for our work.

---

### Meta-Review · Area_Chair_yroJ · 2023-12-12

**Metareview:**

All four reviewers recommend acceptance. The AC agrees with this decision. While the method is incremental in nature (with respect to [Ortiz-Jimenez et al, 2023]), the idea of applying task arithmetic over linearized adapters is interesting and yields good results, as acknowledged by the reviewers.

**Justification For Why Not Higher Score:**

The paper is incremental over [Ortiz-Jimenez et al, 2023]. As pointed out by reviewer HkFi, this work adapts an existing method to the LoRA parameters rather than the original whole model parameters as in [Ortiz-Jimenez et al, 2023].

**Justification For Why Not Lower Score:**

Despite its incremental nature, the proposed method is sensible and yields good results, as noted by all reviewers.

---

### Decision · Program_Chairs · 2024-01-16

Accept (poster)